# Transition Metal Phosphides (TMP) as a Versatile Class of Catalysts for the Hydrodeoxygenation Reaction (HDO) of Oil-Derived Compounds

**DOI:** 10.3390/nano12091435

**Published:** 2022-04-22

**Authors:** Latifa Ibrahim Al-Ali, Omer Elmutasim, Khalid Al Ali, Nirpendra Singh, Kyriaki Polychronopoulou

**Affiliations:** 1Mechanical Engineering Department, Khalifa University, Abu Dhabi 127788, United Arab Emirates; 920020079@ku.ac.ae (L.I.A.-A.); omer.elfaki@ku.ac.ae (O.E.); 2Center for Catalysis and Separations, Khalifa University, Abu Dhabi 127788, United Arab Emirates; khalid.alali@ku.ac.ae; 3Chemical Engineering Department, Khalifa University, Abu Dhabi 127788, United Arab Emirates; 4Physics Department, Khalifa University, Abu Dhabi 127788, United Arab Emirates

**Keywords:** HDO reaction, transition metal phosphides, structure, acidity, characterization

## Abstract

Hydrodeoxygenation (HDO) reaction is a route with much to offer in the conversion and upgrading of bio-oils into fuels; the latter can potentially replace fossil fuels. The catalyst’s design and the feedstock play a critical role in the process metrics (activity, selectivity). Among the different classes of catalysts for the HDO reaction, the transition metal phosphides (TMP), e.g., binary (Ni2P, CoP, WP, MoP) and ternary Fe-Co-P, Fe-Ru-P, are chosen to be discussed in the present review article due to their chameleon type of structural and electronic features giving them superiority compared to the pure metals, apart from their cost advantage. Their active catalytic sites for the HDO reaction are discussed, while particular aspects of their structural, morphological, electronic, and bonding features are presented along with the corresponding characterization technique/tool. The HDO reaction is critically discussed for representative compounds on the TMP surfaces; model compounds from the lignin-derivatives, cellulose derivatives, and fatty acids, such as phenols and furans, are presented, and their reaction mechanisms are explained in terms of TMPs structure, stoichiometry, and reaction conditions. The deactivation of the TMP’s catalysts under HDO conditions is discussed. Insights of the HDO reaction from computational aspects over the TMPs are also presented. Future challenges and directions are proposed to understand the TMP-probe molecule interaction under HDO process conditions and advance the process to a mature level.

## 1. Introduction 

With the world-wide energy demands increasing in an almost uncontrollable manner, the need to produce fuels in an eco-friendly way is imperative for a population that keeps growing. Renewable sources, e.g., wind, biomass, solar, are used to tackle the energy challenges. Using biomass waste to [1,2,3] produce fuels in the absence of carbon dioxide is a promising route to the future energy quota for replacing fossil fuels. By 2030, 32% of energy is expected to be covered by renewable sources [1]. 

Bio-ethanol and biodiesel are the so-called *first-generation bio-diesel*, and they suffer from low efficiency. The *second-generation biofuels* utilize the inedible biomass; in this case, the destruction of lignocellulosic biomass remains a challenge for the process. The bio-fuels derived from cyanobacteria and microalgae are categorized as *third-generation*. 

Liquid and gaseous fuels can be derived from biomass applying different pathways: hydrolysis, pyrolysis and Fischer-Tropsch. Bio-oils derived from the pyrolysis can be subjected to a versatile upgrading procedure, such as zeolite upgrading [4,5,6], steam reforming [7,8,9,10,11,12] and hydrodeoxygenation [13,14] (Figure 1). 

Pyrolysis oil is an attractive renewable fuel used as an alternative to petroleum (fossil) fuel to minimize the environmental impact of the latter. However, the pyrolysis oil contains a high amount of oxygen, between 20% to 40%, which leads to low energy density, high acidity, and thermal and low chemical stability [15,16,17,18]. Therefore, the oxygen should be extracted from the oil; the reaction of hydrodeoxygenation (HDO), specifically the direct deoxygenation method, as described by the reactions below (for the case of fatty acids), can be used to remove the oxygen content in the presence of H_2_ (Figure 2). 

The HDO reaction can happen by two possible approaches: *direct HDO and two-step* HDO. In the first process, unsaturated bonds are hydrogenated using the high pressure of H_2_ (100 bar, ~573 K); separation of hydrogen from green diesel follows. Downstream, green diesel is separated from molecules such as excess hydrogen, water, CO_2_, as well as an amount of C2-C3 alkanes. Dehydration of glycerol (co-product) follows [19]. Higher capital costs are involved in the latter case and lesser hydrogen consumption. The direct HDO has higher H_2_ consumption due to oxygen removal from glycerol backbone towards propane formation. The co-product amount in the second process is much higher than in the first one. In the direct HDO, the electricity consumption has been reported to be higher than the two-step process due to the presence of a compressor for pressurizing the H_2_. Overall, the cost for green diesel manufacturing has been reported to be lower in the two-step process (0.798 USD/kg) than the one step HDO (0.84 USD/kg) for plant operating 0.12 MMT Karanja oil/annum. Assuming a rate of return equal to 8.5%, the minimum selling price of green diesel (1.21 USD/kg, 2-step process) compared to direct HDO (1.186 USD/kg, direct) [14]. The feedstock used brings up to 75% of the production cost [14,19]. Traditionally, metal sulfides (e.g., NiMoS, CoMoS) have been used for the HDO reaction due to their exceptional hydrogenation activity [14,15,16,19,20,21], although they suffer from deactivation in the absence of sulfur (structural deterioration) as is the case for bio-oils [22,23]. Another family of hydrogenation catalysts that are good candidates for HDO reaction are the noble metals (e.g., Pt, Pd, Rh, Ru), although they are costly and exhibit low tolerance towards poisoning in the presence of sulfur (S) and nitrogen (N) compounds [24,25,26]. Recently, transition metal carbides, nitrides, borides and especially phosphide catalysts have become essential in hydrotreatment reactions [27,28,29,30].

In recent years, transition metal phosphides (TMP) have been reported in the open literature as active hydrotreatment catalysts and are associated with increasing demand in the processing of biomass-derived zero-carbon renewable raw materials into fuels and added-value chemicals. TMP are already known catalysts, as they have been extensively used for many catalytic reactions in the energy conversion and catalysis fields, such as HDS [31,32], HDN [33,34], WGS [35,36], photocatalysis [37,38], and HER [39,40]. The superiority of TMPs for the reactions above can be found in particular structural features of the TMPs, such as metal site density at the surface, metal electron density, and BrØnsted acidity. For example, among the S-containing compounds in the HDS reaction are thiol, thioether, thiophene, benzothiophene (BT), dibenzothiophene (DBT), etc. The conventional catalysts for HDS are MoS_2_ and WS_2_ (layered structures) supported on alumina. The metal sites are only located at the edge of the catalysts, introducing an intrinsic limitation in the activity over those catalysts. On the other hand, TMPs with non-layered structure, presented high performances in HDS with Ni_2_P being almost the best HDS catalyst. The structure of TMPs is strongly associated with the methods of preparation [41,42,43]; the latter gives rise to unique electronic properties and coordination/bonding environment. TMPs combine the physical properties of the phosphorus (P) and the transition metal (TM), such as ceramic hardness and strength with electronic properties, such as metal conductivity. At the same time, they have an obvious advantage in terms of cost-effectiveness and environmental-friendliness over the other active metallic phases. 

In the present review article, we focus on the unique structural and electronic properties of the transition metal phosphides (TMPs) and their reactivity towards the HDO reaction. In Section 2, the structural versatility of TMPs is presented with all the different classifications (e.g., compositional, structural, active sites). In Section 3, necessary characterization tools and analytical techniques are critically discussed in terms of enriching our knowledge on structural, compositional, and morphological features of TMPs. Techniques, such as X-ray Photoelectron Spectroscopy (XPS), X-ray Absorption Near Edge Spectroscopy (XANES), Nuclear Magnetic Resonance (NMR), Chemisorption, Transmission Electron Microscopy (TEM) are discussed. Moving to Section 4, the catalytic activity of a plethora of TMPs is discussed for three representative families of bio-oil derivatives with different functionality, namely phenolics, vegetable oils, and furans. As a consequence of the nature of the HDO reaction, deactivation happens; the latter critical issue is discussed in Section 5. To get a deep insight into the TMP surface–probe molecule interactions, computational aspects of the HDO reaction are discussed in Section 6. The last part of this review, Section 7, discusses the outlook and future perspectives of the HDO field using TMPs and the way forward towards the mature knowledge-based design of functional TMPs. 

## 2. A Brief on the Transition Metal Phosphide (TMP)

### 2.1. Structural Concepts

The compounds formed between phosphorus and any d- (e.g., Ni, Mo, W, Co, Fe) or f-metal are called phosphides. Phosphorous can adopt any oxidation state between 0 and 3, resulting in a plethora of structural configurations. The TMPs are refractory metallic compounds exhibiting both metallic and acidic sites [44,45] due to the alloying of TM with P atoms (Figure 3); their classification can be found in Table 1, whereas their physical properties are shown in Table 2. Thorough review articles on the structure of TMPs can be found in the literature [46,47,48,49,50,51,52]. The most frequently used elements in TMPs are iron (Fe), cobalt (Co), nickel (Ni), and molybdenum (Mo). Each TMPs have a variety of morphology and particle size, leading to different properties depending on many crucial parameters, such as the method of preparation followed, P source used, capping agents, temperature, etc. Due to the versatility of their structures (Table 3), TMPs have been used for many catalytic reactions [53]. The catalytic performance of TMPs catalysts, together with the reaction conditions for hydrodeoxygenation reaction of various feedstock are tabulated in Table 4.

Transition metal phosphides (TMPs) have a formula of $M_{x}P_{y}$ which eliminates the electron delocalization around the metal atom due to the slightly higher electronegativity of P than the metal; this leads to electron transfer with a direction from metal to P. The characteristics of the metal phosphide rely on the difference in the M-P electronegativity and the M:P ratio. In terms of bonding, TMPs exhibit a combination of covalent and ionic nature bonds due to the small electronegativity differences that give the metal a small positive charge (δ^+^) and the phosphorus a small negative charge (δ^−^). Therefore, this strong bond leads to high thermal and chemical stability and hardness [54]. 

There are two types of phosphides: (a) phosphorus-rich phosphides (M*x*P*y*, *x* < *y*) and (b) metal-rich phosphides (M*x*P*y*, *x* ≥ *y*) (see Table 1). In particular, the metal-rich catalysts are of great interest as they resemble the metals’ properties (e.g., good thermal conductivity, high thermal and chemical stability) [49]. In the metal-rich phosphides (x:y ≥1) the electrons do not fully surround the phosphorus atom, so intensive M-M interactions can develop. Of particular interest are the metal-rich TMPs as they can present a plethora of crystal structures and compositions (Figure 3). For example, nickel phosphide can be crystallized in Ni_2_P, Ni_5_P_4_, Ni_12_P_5_, Ni_3_P, where Ni_2_P adopts Fe_2_P-type structure [42], whereas Ni_3_P adopts Fe_3_P-type structure [55]. 

Phosphorus-rich (x:y<1) do not have any M-M bonding, particularly when x:y<1/2 because of the poor electrical conductivity. Most of the P atoms can exist in the form of oligomeric chains and clusters and any increase in the content of phosphorus leads to higher reactivity, along with poor thermal stability. As a sequence, at high a temperature, it can decompose to elemental phosphorus and metal-rich phosphides [54]. Especially in the case of alkali metals, alkaline earth metals and some of the transition metals, e.g., zinc phosphides, a large difference in the M-P electronegativity can be noticed and thus high ionicity of M-P bonds ready to hydrolyze. 

**Other TMP classifications:** Another way to categorize the TMPs is to split them into binary, ternary, and supported ones [56]. Different nanostructures of binary TMPs have been reported in the open literature, such as CoP in the form of nanotubes, nanoparticles, nanosheets, nanorods [57,58,59], Co_2_P in the form of nanoflowers, nanoparticles, nanosheets [60,61], Cu_3_P in the form of nanoarrays, nanowires [62,63], and MoP in the form of nanoflakes, nanoparticles [64,65]. 

Ternary phosphides are exceptional catalysts for various reactions, presenting interesting morphologies. Some examples include: NiCo_2_Px (nanowires) [66,67], Ni–Fe–P (nanocubes) [68], CoMoP (core-shell) [69], NiCu-P (porous) [70]. To prepare the above phosphides, a plethora of P sources have been used including NaH_2_PO_2_, TOP, Red P, whereas different methods of synthesis have been followed [71,72]. 

Supported TMPs are famous for increasing the dispersion and stabilizing the active phase (e.g., TMPs); alumina, silica, and carbon are among the most used supports. Additionally, the supports can offer active reaction sites (e.g., Bronsted acidity) and thus participate in the reaction (bi-functional catalysis) [73,74,75]. There is a wide amount of literature covering the use of some supports and the challenges imposed on the process due to their chemistry. For instance, the traditionally industrial support, Alumina, facilitates the Mo-O-Al linkage formation which seems to deactivate part of the active MoP phase, ultimately deteriorating the catalytic activity. Additionally, alumina is subjected to phase transformation (boehmite) when exposed to water (part of HDO reaction conditions). Al_2_O_3_ contains a high water level (30 wt.%), leading to intrinsic oxidation of the metal and the P in the TMPs. Other supports are the mesoporous silica, such as MCM-41 and SBA-15, where both possess high surface area and acid site density. Activated carbons have already been used [73,74,75], but they suffer from the presence of micropores (unsuitable in the HDO reaction) and poor mechanical stability. 

It is well-established that support plays key roles in supported catalysts. On the one hand, the oxygen vacancy sites and surface acidity/basicity might contribute to the activation of reactants and consequently affect products distribution. On the other hand, the support affects the structure of the active phase, such as electronic properties and dispersion. The literature reported that the type of support influences the nature of the metal phosphide phase formed since the oxide support can react with the P precursor salt resulting in the generation of phosphate species [76,77,78,79]. For instance, Alumina interacts favorably with phosphorus-producing ${\mathrm{AlPO}}_{4}$, which prevents the synthesis of ${\mathrm{Ni}}_{2}P$ phase and yields various Ni phases [76,77,78,79].

HDO activity and product selectivity are primarily contingent on the nature of the metal phosphide phase and as well as the type of support. For example, Shi and co-workers [76,77,78,79] synthesized various nickel phosphide catalysts supported on ${\mathrm{TiO}}_{2}$, ${\mathrm{CeO}}_{2}$, ${\mathrm{SiO}}_{2}$, ${\mathrm{Al}}_{2}O_{3}$, HY and SAPO-11. The results indicated that the ${\mathrm{Ni}}_{2}P$ phase was observed on ${\mathrm{TiO}}_{2}$, ${\mathrm{CeO}}_{2}$, ${\mathrm{SiO}}_{2}$ and SAPO-11. The ${\mathrm{Ni}}_{2}P$ and ${\mathrm{Ni}}_{12}P_{5}$ were formed on HY, whereas ${\mathrm{Ni}}_{12}P_{5}$ and ${\mathrm{Ni}}_{3}P_{\;}$ were obtained on ${\mathrm{Al}}_{2}O_{3}$. Clearly, the supported ones exhibited different interactions with P and Ni atoms giving rise to the generation of various ${\mathrm{Ni}}_{x}P_{y}$ phases. The activity for deoxygenation of methyl laurate followed the order: ${\mathrm{Ni}}_{2}P$/${\mathrm{SiO}}_{2}$ > ${\mathrm{Ni}}_{12}P_{5}$-${\mathrm{Ni}}_{3}P_{\;}$/${\mathrm{Al}}_{2}O_{3}$ > ${\mathrm{Ni}}_{2}P$/${\mathrm{TiO}}_{2}$ > ${\mathrm{Ni}}_{2}P$/SAPO-11 > ${\mathrm{Ni}}_{12}P_{5}$-${\mathrm{Ni}}_{2}P_{\;}$/HY > ${\mathrm{Ni}}_{2}P$/${\mathrm{CeO}}_{2}$, and it was revealed that the influence of supports mainly originates from their reducibility and acidity. 

Moon et al. [79] examined the HDO of guaiacol on ${\mathrm{Ni}}_{2}P$ supported on active carbon (AC), ${\mathrm{SiO}}_{2}$ and ${\mathrm{ZrO}}_{2}$ under a temperature of 573 K and 30 atm. Unlike active carbon (AC) and ${\mathrm{ZrO}}_{2}$ support, the in situ XAFS analysis revealed that the ${\mathrm{Ni}}_{2}P$/${\mathrm{SiO}}_{2}$ catalyst experienced oxidation to generate nickel phosphate during the course of reaction. ${\mathrm{Ni}}_{2}P$/${\mathrm{SiO}}_{2}$ has demonstrated higher conversion (87%) than ${\mathrm{Ni}}_{2}P$/${\mathrm{ZrO}}_{2}$ (72%) and ${\mathrm{Ni}}_{2}P$/AC (46%) catalysts.

### 2.2. Functionality (Acidity) of TMPs

TMP surface has a mild acidity, as proved by NH_3_ temperature programmed desorption (TPD) studies [80]. For example, one of the most popular TMP, Ni_2_P, contains not only the metal sites where the hydrogenation takes place but also two types of acid sites, namely: Ni^δ+^ sites and P-OH sites, which are categorized as Lewis and Bronsted acid sites, respectively. Later, how these acid sites participate in the HDO reaction mechanistic steps will be discussed. Unsupported TMPs (e.g., MoP) and supported ones (e.g., Ni_x_P, Co_x_P, W_x_P, Mo_x_P, Fe_x_P and Ru_x_P) are efficient catalysts for HDO since they contain BrØnsted and Lewis acid sites. TM develops a small positive charge $M^{\delta +}$ which act as Lewis acid sites and thus favors reactions such as hydrogenation, hydrogenolysis, and demethylation. BrØnsted acid sites are generated on the P-OH sites that produce active hydrogen species; the hydroxyl groups of P-OH promote hydrogen diffusion (spillover) and contribute to the stability of these species. 

### 2.3. A Close Look on the TMP Active Sites for the HDO Reaction

Two examples of widely used TMP in catalysis are the metal-rich phosphides Ni_2_P and MoP. The crystal structure of MoP is composed of hexagonal layers of P having Mo in the trigonal prismatic positions. All Mo atoms are equivalent; the same applies to the P atoms [81,82]. Two types of metal sites can be found in the Ni_2_P: 4-coordinated distorted tetrahedron and 5-coordinated square pyramidal. The P atoms are located in a face-capped trigonal prismatic environment, though two types of P sites can be found based on the coordination with 4- and 5-coordinated Ni atoms [81,82]. 

Depending on the position of Metal and P in the phosphide lattice, the Ni and P charges can vary. In the case of MoP, Mo and P present only slightly positive and negative charges, whereas, in the case of Ni_2_P, Ni can exhibit either positive or negative charge with P carrying a slightly positive charge [83]. Based on the charge distribution, both phosphides exhibit covalent bonding and metal-like character. 

The so-called *ligand effects* describe the Ni → P charge transfer [84,85]. In addition, P can exert structural effects on the metals leading to an increase in the M-M site distance compared to the pure metal lattice [86]. These effects lower the reactivity compared to the pure transition metals and suppress the phase transitions (e.g., retarding metal sulfides formation in S-containing environments). In turn, P atoms may act as a reservoir of H atoms; the latter can be provided for hydrogenation and hydrogenolysis reactions for the species adsorbed at the metal atoms. 

The bifunctional character of the phosphide catalysts is being constituted by the metal sites (metal atoms or metal-P clusters), and the P-OH groups (acidic sites) at the surface of phosphides due to the presence of strong P-O bonds environment [87]. On the other hand, the selectivity of the hydrogenolysis and hydrogenation reactions have been linked to the population of the OH groups [88]. 

### 2.4. TMPs Preparation Strategies

The TMP can be synthesized following different methods and a plethora of their modifications. Methods, such as solution–phase reactions [89,90], gas–solid reactions [91], and solvothermal reactions [92], have been applied for the preparation of unsupported and supported TMPs. 

*Gas-Solid synthesis:* A typical procedure of applying temperature-programmed reduction (TPR) method in TMPs synthesis has been presented by Cecilia et al. (2009) [93] for the preparation of CoP nanoparticles; Co(OH)_2_ and H_2_PO_3_H were used as the cobalt and phosphorus precursors, respectively. By mixing the two precursors, Cobalt(II) dihydrogenophosphite (Co(HPO_3_H)_2_) was the resultant product. Different supports were used in this study, such as MCM-41 (CoP-10 (Si)), MCM-41 zirconium doped (CoP-10(Zr)), Cab-osil silica (CoP-10 (Cab)), and alumina (CoP-10 (Al)). The TPR step aimed to decompose the above precursor to phosphide in a reducing atmosphere in the 100 to 800 °C range [93]. 

*Liquid phase synthesis:* In another report by Zhang et.al. (2017), CoP nanoparticles were prepared using liquid phase synthesis instead. In this procedure, Co(acac)_2_, 1-octadecene, and TOP were mixed and subjected to heating at 300 °C for 5 h. Following the cooling of the particles, washing was performed using hexane and drying was completed under vacuum at 80 °C [94]. Following the protocol of liquid synthesis, Jiang et al. (2015) [95], synthesized CoP nanoparticles using Co(acac)_2_ as Co sources, whereas Triphenylphosphine (PPh_3_) was used as a P-source in this case. The heating of the precursors in a furnace at 400 °C for 1 h was performed. However, phosphine is a toxic gas so using an alternative is recommended, such as hypophosphites, phosphites, or phosphates. At medium temperatures, hypophosphites decompose (approximately 250 °C), leading to the formation of PH_3_, which subsequently reacts with metal precursor such as metals, metal oxides, hydroxides, salts, and other compounds. In contrast, phosphites or phosphates need hydrogen to create phosphine. Additionally, CoP and Co_2_P nanoparticles were produced by Ha et. al. (2011) [96] using the solution phase arrested precipitation. 

Lu et al. have reported on the controlled synthesis of Ni_x_P_y_ nanoparticles encapsulated by silica following solution-based synthesis in oleylamine (OAm) with trioctylphine (TOP); the method is anticipated to effectively control the morphology and phase of MP nanoparticles due to the high boiling point and ease in controlling the metals’ complexity property. For example, OAm can have a triple role as a surfactant agent, a solvent medium and a reducing agent and stabilizer for the synthesis of MP nanoparticles. Additionally, TOP acts as the P source, surfactant and solvent. There are two stages for a typical synthesis of ${\mathrm{Ni}}_{2}P$ nanoparticles in OAm where the first stage produces Ni nanoparticles, and in the second stage, ${\mathrm{Ni}}_{2}P$ is formed. In the first stage, the Ni nanoparticles are formed by decomposition of Ni precursor such as $\mathrm{Ni}{\left(\mathrm{acac}\right)}_{2}$ above 200 °C. During the second stage, P insertion is taking place to produce ${\mathrm{Ni}}_{2}P$ above 300 °C. The ultrafine Ni_2_P nanoparticles were confined in mesoporous silica so to design a sintering-resistant catalyst for the catalytic reduction of ${\mathrm{SO}}_{2}$ to sulfur in the $H_{2}$ presence [89,90]. 

D’Accriscio et al. reported on the synthesis of Fe_x_P nanoparticles using tetrakis(acyl)cyclotetraphosphane, P_4_(MesCO)_4_; the latter P source is a very stable one and not air-sensitive like TOP or white phosphorous [97] (Figure 4a). At a temperature as low as 250 °C, they reported the synthesis of FeP and Fe_2_P nanoparticles. The local order in the materials was studied by using XPS and atomic pair distribution function (PDF). The authors reported that crystalline FeP is formed via an intermediate amorphous one. The FeP nanoparticles exhibited a spherical hollow shape of 8.9 nm size. TEM studies on the Fe_2_P showed that they had spherical hollow shapes with two shells with the inner shell being about 1 nm thickness (Figure 4a). Huiming Li et al. [98] reported on a systematic effort to synthesize nanocrystals of Ni-P with different stoichiometries (Ni_5_P_4_, Ni_2_P, Ni_12_P_5_) by adjusting the P/Ni precursor ratio, the temperature and the reaction duration (Figure 4b) and starting from OAm and TOP. Ni_5_P_4_ monodisperse with a size of 5.6 nm was achieved; such nanocrystals, due to the high surface areas, are anticipated to exhibit promising catalytic activity. Shanfu Sun et al. [99] **(**Figure 4c) synthesize (Ni_x_Fe_1−x_)_2_P using a two-step process. Starting from a NiFe foam which was subjected to corrosion, they succeeded in growing NiFe-LDH nanosheets; the latter were subsequently phosphorized using NaH_2_PO_2_ under an inert atmosphere (N_2_). 

An example of ternary phosphide synthesis has been reported by Ye et al. [100], who synthesized Co_2−x_Fe_x_P using TOP and OAm. It was reported that the synthesis temperature had a crucial role to adjust the morphology of the resultant phosphide; Co_1_._5_Fe_0_._5_P (rice-shaped) or Co_1_._7_Fe_0_._3_P (split). In another study by Lawes et al. [101], the synthesis of the Co_x_Fe_2−x_P in the compositional range 0 < x < 2 was investigated. In the first step of the process, starting from carbonyl precursors of both metals, Co-Fe alloy nanoparticles were prepared, which were subjected to reaction with the TOP in the 330–350 °C temperature range. The authors noticed that the Co-rich compositions prefer to crystallize at low temperatures, whereas the Fe-rich follow an opposite temperature profile. At intermediate compositions (values of x), the predominant growth mechanism is aggregation. Another procedure to synthesize (Co_x_Fe_1−x_)_2_P is through the synthesis of the relevant ferrite (CoFe_2_O_4_), which is subjected into reaction with TOP [102]. Many different compositions of ternary TMPs have been reported, such as Ni-Co-P, Fe-Ni-P, Fe-Mn-P, Co-Mn-P, Co-Rh-P, and Ni-Ru-P [103]. It is known that their synthesis is rather demanding due to the combination of the reactivity from two metal compounds, whereas manipulating the stoichiometry is important as the latter can change their catalytic chemistry of them. The content of the metals in the synthesis solution frequently happens does not coincide with the stoichiometry of the final phosphide. That is the reason behind the strategies followed to synthesis of the ternary phosphides. Usually, the bimetallic or the oxide are prepared and their reaction with the P source follows (Figure 5). 

**Table 4 nanomaterials-12-01435-t004:** The catalytic performance of different TMPs catalysts used in deoxygenation reactions of various reactants.

Catalysts	Reactants	Reaction Conditions		Coversion (%)	Selectivtiy (%)	Reference
		T (°C)	P (bar)			
$\mathrm{Pd}-{\mathrm{Ni}}_{2}P\;$ $/{\mathrm{SiO}}_{2}$	Phenol	220	20	~100	~92 Cyclohexane	[104]
MoP $/{\mathrm{SiO}}_{2}$	2-methyltetrahydrofuran	275	1	15	14 Pentane 70 Pentenes 7 Pentadienes	[105]
$\mathrm{CoP}/{\mathrm{SiO}}_{2}$				11	40 Pentane 15 Pentenes 25 C14	
${\mathrm{Ni}}_{2}P\;$ $/{\mathrm{SiO}}_{2}$				12	67 pentane 2 pentanone 15 C14	[106]
CoP $/{\mathrm{SiO}}_{2}$	dibenzofuran	275	30	~90	~72 Bicyclohexane	[107]
** ${\mathrm{Ni}}_{2}P$ **	guaiacol	300	1	92.9	14.1 Phenol 58.7 Benzene	[108]
** ${\mathrm{Ni}}_{12}P_{5}$ **				93.9	31.5 Phenol 34.9 Benzene	
** ${\mathrm{Ni}}_{3}P$ **				99.8	23.9 Phenol 22.8 Benzene	
${\mathrm{Ni}}_{2}P$/ pillared -ZSM-5		260	40	78	~82 cyclohexane	[109]
FeMoP	Anisole	400	21	>99	90 Benzene 10 Cyclohexane	[110]
${\mathrm{Ni}}_{3}P$ $-\;{\mathrm{Ni}}_{12}P_{5}$ $/\gamma -{\mathrm{Al}}_{2}O_{3}$	methyl laurate	340	20	~98	~96 C11 + C12~12 C11/C12	[111]
${\mathrm{Ni}}_{2}P\;$$\mathrm{supported}\;\mathrm{on}\;{\mathrm{SiO}}_{2}$, MCM-41		340	20-30	97-99	~100 C11 + C12	[112]
${\mathrm{Ni}}_{2}P$/ SBA-15	methyl oleate	270	30	~84	~45 C18/(C17+C18)	[113]
${\mathrm{Ni}}_{2}P$ $/{\mathrm{SiO}}_{2}$	methyl laurate	340	30	~97	~87 C11 + ~12 C12	[114]
$\mathrm{MoP}/{\mathrm{SiO}}_{2}$		340	30	~90	~4 C11 + ~80 C12	
$\mathrm{NiMoP}/{\mathrm{SiO}}_{2}$		340	30	~98	~51 C11 + ~49 C12	
${\mathrm{Ni}}_{1.5}P$/activated carbon	palmitic acid	350	1	100	57 C15 7 C11-C14 21 alkenes	[115]
${\mathrm{Ni}}_{x}P\;$/HZSM-22		350	1	99.6	-	[116]
$10\;\mathrm{wt}\%\;\mathrm{CoP}/\gamma -{\mathrm{Al}}_{2}O_{3}$	2-furyl methyl ketone	400	1	100	~100 Methyl Cyclopentane	[117]

## 3. Characterization of TMP Features towards Unveiling Outstanding Performance 

### 3.1. Structural Features (XRD, Raman, TEM, SAED)

To understand the structure–property correlation for the TMP, X-ray diffraction (XRD) is one of the primary techniques to be utilized to be able to assess the crystallinity of the TMP nanostructures. Along the same lines, Selected Area Electron Diffraction (SAED) collected in the TEM analysis provides complementary information with XRD. XRD has been extensively used to validate the success of synthesis of TMPs, e.g., CoP/SiO_2_ prepared using the TPR method [105]. Using XRD analysis after the TPR synthesis can advocate the reduction process efficiency and conditions, the phenomena involved, and the suitability of the preparation method for the scale up of the process. In some cases, fluoresce can happen, deteriorating the quality of the XRD pattern, e.g., the case of FeP, though the presence of FeP could still be identified. Following a TPR synthesis protocol on different P precursors (phosphite [HPO_3_]^2−^ vs. phosphate [PO_4_]^3−^) can be challenging for the TMPs’ quality (absence of impurity phases) and XRD can offer an assessment tool. Crystallinity of the TMP can also be different based on the nature of the metal (e.g., better quality crystals are formed for CoP vs. WP, as the latter starts at a higher oxidation state) [105]. The presence of pure metallic phases as impurities have also been reported [105]. The crystallite size of TMPs can be calculated using the Scherrer formula. An example is presented by H. Y. Zhao et al. [108], who studied supported TMPs, such as MoP/SiO_2_, Ni_2_P/SiO_2_, WP/SiO_2_ and Co_2_P/SiO_2_. Among the catalysts studied, Ni and Mo catalysts had the smallest particle size. In the case of MoP, no XRD peaks were detected, indicating the high dispersion of the MoP phase on silica and small particle size (less than 5nm). In the case of Ni_2_P, a broad XRD pattern was received, showing the small crystallites size. Additionally, nickel phosphide showed three peaks that line up with the iron pattern but with more narrow peaks, which means Fe has a larger particle than Ni.

Advanced diffraction techniques, such as in situ XRD, have been employed to monitor the TMP phase evolution; Inocencio et al. [118] showed that this technique could shed light on the preparation mechanism and phase transformations that are taking place. The method of preparation followed was for TPR of phosphates (Ni_x_P_y_O_z_; Mo_x_P_y_O_z_; Co_x_P_y_O_z_ and Fe_x_P_y_O_z_, two-step synthesis). The presence of Ni_2_P, MoP, CoP/CoP_2_, FeP and WP was identified using in situ XRD. Under the reduction atmosphere, the phosphate precursor was decomposed towards the formation of metal oxide (Mo case) or pyrophosphate phases (Ni, Fe) with no formation of metallic phase. In an attempt to understand the TPR over phosphates, in the case of Mo_x_P_y_O_z_, two peaks at 437 and 554 °C are exhibited and this finding is in agreement with a previous study by Bui et al. [105]. Based on the in situ studies, the Mo(OH)_3_PO_4_ phase is initially formed and then likely decomposed to MoO_3_ and MoOPO_4_ and some amorphous portion. The CoP_x_O_y_ TPR presented a peak in the 500–530 °C range and a broad peak at 580 °C reflecting the reduction of Co-phosphate to metal and PH_3_, which reacts with the metal towards CoP formation. Peroni et al. [119] studied the catalyst’s particles sizes either using X-ray or statistical analysis of TEM micrographs (see table below). From TEM results, MoP catalysts tend to have smaller particles, i.e., 29.2 nm compared to the ${\mathrm{Ni}}_{2}P$ particles, i.e., 132.4 nm. However, the low temperature methods tend to have a smaller particle size than the TPR approach. In addition, the citric acid that is used during the synthesis reduced the Ni_2_P crystal size from 132.2 nm to 49 nm because of the suppression of the sintering of active phases. In addition, Raman spectroscopy was used to study ternary solid solutions phosphides of Fe_1-x_Ni_x_P (0 < X < 0.3) composition. Raman spectroscopy showed that variation of Ni content in the solid solution caused an almost linear shift of several peaks. Experimental peak positions were found to match with the phonon frequencies, giving a better understanding of the frequency-composition dependence [120]. 

### 3.2. Dispersion of the TMPs over a Support

There are many reports where the TMPs are dispersed on a high surface area support, preferably with acid sites so as to modify the ultimate acidity of the final structure and increase the active phase (TMP) exposure and bifunctionality of the catalyst. Such supports can be MCM-41, SBA-15, other types of silica, alumina, and carbon. In the cases of supported TMPs, the dispersion of TMP needs to be evaluated for appropriate assessment of the catalytic activity. For evaluating the dispersion, CO chemisorption [94] along with CO infrared studies are reported. CO is a typical molecule used for such studies due to its small size and its C-O stretching frequency, which is well-defined for the case where CO adsorption takes place on metal surfaces [121,122]. Based on the CO uptake values (µmols/g), blockage of active metal sites can be predicted and correlated with the synthesis conditions followed. It has been observed in the case of the phosphate method where the P^5+^ is not easily reducible and higher temperatures are needed to convert into phosphide that unreduced phosphate stays bound on the metal, reducing the active sites [123]. In addition, smaller crystallite size corroborates with a higher dispersion, giving rise to higher CO uptakes (µmols/g). 

Infrared spectra (IR) of CO adsorbed on the surface of reduced Ni_2_P/SiO_2_ catalysts gives rise to four species, as follows: (1) CO adsorption on atop Ni sites, (2) CO adsorption on bridge Ni sites, (3) formation of adsorbed species Ni(CO)_4_, and (4) formation of adsorbed P=C=O species on the surface, terminal attachment of CO species to P [121]. The relative CO site densities and the ∆*H*_ads_(CO) are reported. In the case of MoP [124], CO chemisorption on different Mo sites is shown in Figure 6. 

### 3.3. Surface Studies and Coordination Environment (XPS, XANES, NMR)

The surface chemistry of TMPs can be studied using X-ray photoelectron spectroscopy (XPS). A closer look at the binding energies (BE) can shed light on the charge transfer as well as the hybridization of phosphorous. In general, in the case of P 2p, the peak at 129.5 and 130.5 eV assigned to P 2p_1/2_ and P 2p_3/2_ are the fingerprint of TMP. Other peaks originating from P have been reported at 132.5, 133.6, 134.2, and 135.5 and are linked to P-C, P-O, P-O-M, and metaphosphates [125,126]. In an attempt to fully utilize the XPS data, someone has to have a careful look at the main and satellite peaks of the metal as well. Usually, positive shifts of BE on the metal can imply a partial positive charge on the metal site, e.g., M^δ+^; the latter is usually accompanied by a negative shift of BE for P, attributing a partial negative charge to the latter (P^δ-^) and dictating the direction of charge transfer. 

A compile of XPS as a tool for TMPs studies is discussed in what follows. Burns et.al. (2008), [127] used XPS spectra to study the electronic environment of the Co_2_P/SiO_2_ catalyst. The peak at BE 781.6 eV is linked to Co^2+^ and the one at BE 134 eV to P^5+^. Those two peaks are originated due to the passivation step that was completed at the end of the synthesis. Based on the BE 778 and 129.2 eV, which are associated with elemental cobalt and phosphorus, respectively, it can be stated that Co is positively partially charged, and the P is negatively partially charged, and this corroborates the Co → P charge transfer. In the same study, CoP/SiO_2_ and Co_2_P/SiO_2_ are also presented. The relative intensities of Co and P in the Co_2_P are opposite compared to the ones of CoP, which indicates that CoP has a P-rich passivation layer. In addition, the peak shift by 0.8 eV at for CoP to higher energies indicates that there is higher charge transfer from cobalt to phosphorus in CoP than in Co_2_P [127]. 

In their study of optimizing the synthesis of TMPs starting with phosphates and phosphites and following a controlled reduction environment, Bui et al. [94] reported a detailed XPS study of Ni_2_P and WP. In the case of Ni_2_P, a P peak at 128.6 eV (P 2p_3/2_) is the main proof for the Ni_2_P phase formation [128], while the P 2p_3/2_ at higher BE (135.0 eV) was assigned to PO_4_^3−^ (as a result of passivation of the phosphide layer). In the case of the phosphite method of synthesis, unreduced phosphites can also give rise to a P 2p_3/2_ at 133.6 eV. So, this reveals another benefit of using XPS: assessing the purity of the TMP. Similar peaks were observed for the WP. 

Near-Edge X-ray absorption fine structure (NEXAFS), also known as XANES, is a great tool to probe parameters, such as chemical bonding, electronic structure, and interactions in the TMPs. The collected XANES spectra are usually fitted with standard compounds. Yun et al. [14] used EXAFS due to the XRD limitations imposed by the significantly small Ni_2_P size (3 nm), making it undetectable in XRD. The FT spectrum shows two peaks at 0.180 and 0.228 nm, which are assigned to the Ni-P and Ni-Ni chemical interaction [129]. A three-shell curve fitting was followed, leading to Ni-P(I), Ni-P(II), and Ni-Ni bonds lengths of 0.2207, 0.2344 and 0.2551 nm, which are associated with 1.8, 2.8, and 2.0 coordination numbers. The data allow us to assess that Ni-Ni coordination is smaller, whereas Ni-P is higher in small Ni_2_P crystallites. 

Transition metal phosphides possess versatile electronic structures and properties that are anticipated to give rise to unique signals in the ^31^P NMR, including difficulties in tracing the signal or the impossibility to carry out such characterization. NMR spectra features can unveil the characteristics of the TMPs electronic structures. This is showcased in the rigorous research conducted by Bekaert et al. [130] on a series of V, Fe, Co, and Ni phosphides, combining ^31^P MAS NMR and first-principles calculations. The study shows that the electronic structure of the TMPs (diamagnetic, paramagnetic, metallic) and the ^31^P NMR signal are correlated. More attention needs to be paid in the case of paramagnetic TMPs (e.g., Co_2_P), where shorter relaxation delays, a decrease in the signal, and line broadening, can lead to a perplex spectrum. In the case of Co_2_P, short Co-Co distances lead to a strong Co-Co bond interaction giving rise to metallic conductivity and high values for NMR shifts; this amount of high shift makes it possible to differentiate between phosphides and phosphates [131]. In addition, ssNMR nanocrystallography has been used for facets analysis of TMP [132]. In particular, ^31^P DFT-assisted solid-state NMR coupled with HRTEM and XRD were used to evaluate the exposed crystal facets of the terminal surfaces in the case of Ni_2_P nanoparticles. Results showed that two facets (0001-A) and $\left(10\bar{1}0\right)$ are the predominant; the former is the major one. These exposed facets cause a shift in the Ni d-band towards the Fermi level, enhancing the catalytic activity. 

Wanmolee et al. [133] monitored the phase transformation of ${\mathrm{CuHPO}}_{4}.H_{2}O$ precursor into ${\mathrm{Cu}}_{3}P$ phase using in situ XAS and XRD characterization techniques. The atomic coordination and electronic structure were investigated via EXAFS and XANES. The $H_{2}$-TPR was performed in situ for the calcined catalyst by collecting the in situ XAS spectrum at the temperature range of 25–650 °C. According to the XANES results, at temperatures below 110 ºC, the divalent copper ions (${\mathrm{Cu}}^{2+}$), in ${\mathrm{CuHPO}}_{4}.H_{2}O$ precursor, were predominantly observed. Notably, successive dehydration of water from the precursor is observed with a temperature rise up to 200 °C; thereby, the intermediate precursor ${\mathrm{Cu}}_{2}P_{2}O_{7}$ is formed. Above 250 °C, the ${\mathrm{Cu}}^{2+}$ species completely disappeared, while the ${\mathrm{Cu}}^{0}$ and ${\mathrm{Cu}}^{1+}$ species became prevalent, signifying the formation of metal Cu and $\mathrm{Cu}{\left({\mathrm{PO}}_{3}\right)}_{2}$. Above 320 °C, the ${\mathrm{Cu}}_{3}P$ phase is formed which was borne out by the predominance of ${\mathrm{Cu}}^{1+}$ species. Interestingly, the ${\mathrm{Cu}}_{3}P$ phase experienced various transformations in the space group, as conformed by XRD analysis; however, the crystal structure was finally relaxed to $P6_{3}\mathrm{cm}$ space group following the cooling down. The FT-EXAFS revealed that copper atoms are thoroughly bonded with the phosphorus atoms with Cu-Cu radial length of 2.18 Å. 

## 4. TMPs as Catalysts for the Upgrading of Bio-Oil through HDO Reaction

Hydrodeoxygenation (HDO) reaction is used to remove the oxygen in bio-oil without losing carbon (C) and hydrogen (H) atoms. The commercial catalysts, such as Ni/Co-MoS_2_ supported on alumina, suffer from oxygen-induced deterioration in the absence of H_2_S, leading to the ultimate deactivation of the catalysts. Moreover, the acidic sites of alumina tend to favor the coke deposition and deactivation of the catalysts. TMPs are suggested as thermally and chemically stable catalysts for the HDO reaction and as good alternatives to noble metal catalysts. Each TMP may have a different reaction path leading to different products. The HDO mechanism over TMPs and noble metals are compared in Figure 7 and Figure 8; the former is more advantageous cost-wise and selectivity-wise (less prone to secondary reactions, such as C-C bond hydrogenolysis and methanation). The benefit of using TMPs can be summarized as follows: for instance, in the case of popular Ni_2_P, Ni contains a slight positive charge and phosphorus has a slight negative charge because of the partial transfer of electron density from nickel to phosphorus (ligand effect). The metal-rich phosphide catalysts contain metal sites (responsible for hydrogenation and hydrogenolysis) and two types of acid sites (Lewis acid sites, Br∅nsted acid sites). Ligand effect is responsible for the Lewis acid sites (${\mathrm{Ni}}^{+\delta }$), while the Br∅nsted acid sites (P-OH groups) exist on the surface of phosphide. In brief, the advantage of TMPs is the *ligand effect (electronic)* and the *ensemble effect (geometric)* [134]. 

### 4.1. Lignin Phenolic Derivatives

Lignin is a natural polymeric material composed of the monomeric units syringyl (S), guaiacyl (G), and p-hydroxyphenyl (H). Lignin phenolic derivatives have been studied as products of bio-oil processing. TMPs have been extensively used for the HDO reaction of model compounds of this category, such as phenol [139], guaiacol [140], catechol [141], and cresols [142]. 

Under realistic HDO conditions, a temperature range of 380–430 °C and high H_2_ pressure is used. Many reactions, including cleavage of interunit linkages, deoxygenation reaction, ring hydrogenation reaction, as well as the removal of alkyl and methoxyl moieties, occur. The hydrogen pressure, typically 50–150 bar, strongly influences the oil yield. Catalysts design criteria can be as follows: (a) high activity for hydrogenolysis and/or cleavage of C-O-C and C-C bonds (metal sites function), (b) low activity for ring hydrogenation; (c) good selectivity for aromatic compounds (d) tolerance against coking and sulfur. Another critical parameter is the active metal particle size; the latter regulates the C-O cleavage activity and tunes the selectivity to aromatic compounds. For example, tiny metal particles suppress the co-adsorption of the aromatic ring and H_2_ and thus the hydrogenation of the first. Apart from the metal particle size, the geometry of adsorption is also a crucial factor. For example, it has been reported that using Pt(111) surface favors the anisole horizontal orientation (parallel to the surface), while adding Zn to the Pt(111) can induce a tilted mode of adsorption. These two completely different geometries of adsorption favor the ring saturation and the selectivity to aromatic compounds, respectively [143]. 

On a comparative note, in a metal-supported catalyst, the metal sites provide the function of hydrogenation, whereas the support provides the oxophilic sites; the latter tune the C-O cleavage activity of the catalyst. Popular oxophilic metal oxides with strong Lewis acid character are the VOx, NbOx, MoOx, ReOx, WOx, and TaOx. However, studies by Jing 2019b [126] showed that the catalytic activity does not coincide with the Lewis acidity order. 

Trying to get an insight and consider phenol as the model compound of this category, there are three different mechanistic pathways for removing -OH in the phenolic ring. In particular, these routes are: (a) direct hydrogenolysis of -OH in the phenol ring, (b) hydrogenation followed by dehydration and then dehydrogenation, and (c) tautomerization followed by hydrogenation and dehydration route [143]. 

The main focus of the research in this front is to design selective catalysts that allow the lowering of the use of hydrogen while preserving the aromatic character of the lignin compound. HDO of lignin model compounds can be efficiently performed over a copper chromite catalyst (Deutsch and Shanks, 2012). For the benzyl alcohol, the hydroxymethyl group is very reactive under HDO. In the case of anisole, demethoxylation is the primary reaction pathway compared to demethylation and transalkylation. The latter is predominantly for conventional hydrotreating catalysts. The hydroxyl group of phenol contributed to the activation of the aromatic ring toward cyclohexanol and cyclohexane production. 

Guaiacol is another popular model compound of pyrolysis oil with a challenging molecule to fully deoxygenate due to the co-presence of phenolic and methoxy groups. In their study, Zhao et al. [144] reported that HDO of guaiacol leads to benzene and phenol as the major products with a small amount of methoxybenzene. Moreover, the formation of reaction intermediates, such as catechol and cresol, were traced. Guaiacol (methoxyphenol) is used due to the high lignin concentration that tend to be highly coke. In the same study, the performed comparison of the TMP catalysts with two commercial ones is noteworthy to be discussed. In particular, 5% Pd/Al_2_O_3_ catalyst activity was higher at lower contact time, but catechol was the sole product. A sulfide catalyst, commercial CoMoS/Al_2_O_3_, had low activity for the HDO of guaiacol under the studied conditions. 

The TOF was found to follow the order Ni_2_P/SiO_2_ > Co_2_P/SiO_2_ > FeP/SiO_2_ > WP/SiO_2_ > MoP/SiO_2_. The nickel and cobalt TMPs had a smooth activity which is an indication of no deactivation. However, Fe has a high variation of TOF with a small range of temperature, which means it underwent deactivation, which may occur due to coking. These results propose Ni and Co as the most reactive and stable TMP in the HDO of guaiacol. 

The reaction network of HDO of guaiacol HDO involves the hydrogenolysis of the methyl–oxygen bond (O-CH_3_ group) leading to catechol formation, followed by the hydroxyl groups removal towards phenol production and water and then hydrocarbons (benzene or cyclohexene) [145,146,147]. Activation energies for the overall reaction of guaiacol were found to be in the 23–63 kJ/mol range; these values were lower compared to a network involving carbon-oxygen bond breaking (>240 kJ/mol) [144]. These low activation energies suggest that another pathway is followed for the oxygen removal: that of hydrogenation of aromatic double bonds followed by water elimination [145,146,147]. Recently, another mechanistic scheme has been proposed for the HDO of guaiacol, eliminating the production of catechol through the direct hydrogenolysis leading to phenol production from guaiacol [148]. 

In another study, Inocencio et al. [118] (Figure 9) investigated the HDO performance of TMPs (Ni, Mo, Co, Fe, W) for the phenol HDO reaction. The highest activity was found over the FeP, whereas the order of activity was FeP > MoP = Ni_2_P > WP = CoP. The TOF (min^−1^) was found to be 1.94, 0.67, 0.28 and 0.44, respectively. Benzene was the main product for all the under-study catalysts, along with cyclohexene, cyclohexane, cyclohexanone, and C5-C6 hydrocarbons.

Inocencio attempted to correlate the measured activity results with the structure of the TMPs used. In particular, the crystal structures for the different TMPs are: MoP (WC type hexagonal); Ni_2_P (Fe_2_P-type Orthorhombic); WP, CoP, FeP (MnP-type Orthohombic). It was found that the distribution of products depends on the type of TMP crystal structure. The direct deoxygenation reaction is running over WC (hexagonal) and Fe_2_P (Orthorhombic), whereas MnP (Orthorhombic) structure seems to facilitate the aromatic ring hydrogenation. The following selectivity order was found: Ni_2_P > MoP > CoP ≈ WP > FeP. This result has to be seen associated with the P/Metal ratio for each crystal structures. WC (hexagonal) and Fe_2_P (Orthorhombic) exhibit a higher phosphorus/metal ratio than MnP (Orthorhombic), which promotes stronger adsorption of the model molecule on the Ni cation through the oxygen atom. Gonçalves et al. [149] compared the activity of Ni and Ni_2_P supported catalysts on SiO_2_ and ZrO_2_ for the direct deoxygenation of m-cresol to toluene. Among the catalysts investigated, the Ni_2_P/ZrO_2_ had the highest activity. It was also shown that the support nature (SiO_2_, ZrO_2_) did not impact the direct deoxygenation route. The acidic nature of zirconia was found to affect the aromatic ring or double bond hydrogenation along with the dehydration and isomerization reactions. 

### 4.2. Vegetable Oils

Triglycerides of higher fatty acids are the main constituents of vegetable oils. Typically, vegetable oils contain triglycerides of palmitic acid (C16:0), oleic acid (C18:0), and linoleic acid (C18:2) [143]. The carbon chain length in the acids resembles the length of the carbon backbone of the diesel fuel and this makes the vegetable oils an ideal candidate for green diesel production. There is extensive literature on metal-supported catalysts for the HDO reaction of the higher fatty acids [9,10]. Shi et al. [150] studied supported Ni_2_P catalyst on SiO_2_, CeO_2_, TiO_2_, SAPO-11, γ-Al_2_O_3_, and HY zeolite, with a Ni/P ratio of 1.0 for the HDO reaction of methyl laurate to yield C11 and C12 hydrocarbons. It was found that Ni_2_P/SiO_2_ had the highest conversion rate at all temperatures, which related to the SiO_2_ support effect (e.g., reduced acidity or reducibility), demonstrating the vital support role on the TMPs activity. Activity and selectivity were both found to increase with temperature.

M. Peroni et al. [151] (Figure 10) studied different TMPs (WP, MoP, and Ni2P) for the catalytic hydrodeoxygenation of palmitic acid. The exposed metal sites are directly correlated with the catalytic activity. The activation energies were found to be in the 57 kJ mol^−1^ for MoP to 142 kJ mol^−1^ for WP. On WP, the conversion of palmitic acid proceeds via R-CH_2_COOH → R-CH_2_CHO → R-CH_2_CH_2_OH → R-CHCH_2_ → R-CH_2_CH_3_ (hydrodeoxygenation). Decarbonylation of the intermediate aldehyde (R-CH_2_COOH → R-CH_2_CHO → R-CH_3_) was a crucial intermediate in the case of MoP and Ni_2_P. An alternative pathway was followed on Ni_2_P: that of dehydration to a ketene, followed by decarbonylation. The important role of Lewis acid sites was revealed as the alcohol dehydration (R-CH_2_CH_2_OH → R-CHCH_2_) was found to depend on their concentration.

### 4.3. Cellullose and Semi-Cellullose Furan Derivatives

Acid-catalyzed transformation of cellulose leads to the formation of furanic derivatives and by-products because of the acid process conditions. The HDO reaction of furfural [152], 2-methylfuran [153], valerolactone [19], and levulinic acid [154] has been investigated over TMPs. 

Bui et al. [105] studied the TMPs catalytic activity for HDO of 2-Methyltetrahydrofuran (2-MTHF); the TMPs were prepared following phosphite and phosphate method. Catalysts prepared by phosphide method (I) have a slight difference at TOF ($s^{-1}$) with an order of ${\mathrm{Ni}}_{2}P/{\mathrm{SiO}}_{2}>\mathrm{CoP}/{\mathrm{SiO}}_{2}=\mathrm{MoP}/{\mathrm{SiO}}_{2}\sim \mathrm{WP}/{\mathrm{SiO}}_{2}$ with expectation of FeP(I) that has a very low activity. On the other hand, the phosphate method catalysts (A) have different activity following ${\mathrm{Ni}}_{2}P/{\mathrm{SiO}}_{2}>\mathrm{WP}/{\mathrm{SiO}}_{2}>\mathrm{CoP}/{\mathrm{SiO}}_{2}\sim \mathrm{MoP}/{\mathrm{SiO}}_{2}>\mathrm{FeP}/{\mathrm{SiO}}_{2}$. As observed from the results, ${\mathrm{Ni}}_{2}P/{\mathrm{SiO}}_{2}$ is the most effective catalyst while $\mathrm{FeP}/{\mathrm{SiO}}_{2}$ is the lowest one. In addition, CoP and MoP are more active using the phosphite method (I) while for ${\mathrm{Ni}}_{2}P,\;\mathrm{WP}\;\mathrm{and}\;\mathrm{FeP}$ is the opposite where ${\mathrm{Ni}}_{2}P\;\left(A\right)$ is 1.4 times than ${\mathrm{Ni}}_{2}P\;\left(I\right)$. The difference between phosphate and phosphite results are not significant so using the phosphite method is more practical since it does not require calcination whereas use of lower reduction temperature preserves porosity. The same results were found at a lower temperature (275 °C) where the order of activity of the catalyst by phosphite method is $\mathrm{WP}>{\mathrm{Ni}}_{2}P>\mathrm{MoP}>\mathrm{CoP}>\mathrm{FeP}$ and by phosphate method is ${\mathrm{Ni}}_{2}P>\mathrm{WP}>\mathrm{MoP}>\mathrm{CoP}>\mathrm{FeP}$. However, the phosphite method can affect the HDO reaction pathway caused by the deposition of excess phosphorus on the surface. In all cases, the phosphide method has higher conversions except for ${\mathrm{Ni}}_{2}P\;\mathrm{and}\;\mathrm{FeP}$ and needs further studies. The average total conversions of the catalysts are ${\mathrm{Ni}}_{2}P\left(16\%\right)>\mathrm{WP}\left(13.5\%\right)>\mathrm{MoP}\left(5.5\%\right)>\mathrm{CoP}\left(3\%\right)>\mathrm{FeP}\left(0.7\%\right)$. In addition, the study shows that the catalyst did not have any activity at 275 °C  and deactivate at 300 °C therefore the apparent activation energy $E_{a}$ calculated at range of 300-325 °C. The activation energy range was between 88–137 kJ/mol which is lower than the C-O bond dissociation energy (360 kJ/mol). The average HDO conversions of catalysts in order are $\mathrm{MoP}>\mathrm{FeP}>\mathrm{WP}>{\mathrm{Ni}}_{2}P>\mathrm{CoP}$. Fe, Co, Ni and Pd generate C4 products by decarbonylation mechanism while Mo and W do not. MoP (I) and (A) has the highest HDO activity but not the highest in the total conversion. 

The HDO of furfural towards the production of 2-methylfuran (MF) was investigated over different TMPs (Figure 11 and Figure 12), having as a particular focus of the study to establish the role of P and M/P stoichiometry in the TMP on the catalytic activity. Furfural is an important chemical in the resins industry. Metal-based catalysts (e.g., Co [155], Pt [156], Pd [157] are somewhat too active in their interaction with the furan ring leading to the ring opening or ring hydrogenation reactions. This issue seems to be efficiently addressed using TMPs. With a closer look at the furan ring-metal (Ni or NiP) interaction, the following remarks can be made: the interaction involved d-band of Ni and π* of furan ring. In the case of Ni/SiO_2_ catalyst, Ni holds most of the electronic density, making the interaction with the furan ring severe (opening). As P is introduced, Ni electronic density drops and the d-π interaction between Ni-P and furan ring becomes milder. The authors performed furfural infrared spectroscopy (IR) studies where the bands at 1675 and 1695 cm^−1^ were both assigned to the C=O stretching vibration of the carbonyl group, whereas also bands corresponding to the C=C stretching vibration coming from the furan ring were identified. It is interesting to note that as the P increases (P/Ni ratio) the carbonyl bands were found to shift to smaller frequency; namely from 1675 (Ni/SiO_2_ case) to 1655 cm^−1^ (Ni_2_P/SiO_2_ case); the latter corroborates for a boost in the carbonyl-surface interaction with the P increase. Regarding the planar adsorption geometry of the furan ring, this seems to be disturbed by the P increase in the TMP. The authors also provided another explanation on the role of P as BrØnsted acid site in the catalytic activity. As it is known that BrØnstead sites close to metal sites favor the C-O hydrogenolysis reaction by protonating the -OH or -OCH_3_ group, similarly it is inferred that P-OH sites can facilitate the C=O bond hydrogenolysis. 

Another member of the furans family, benzofuran, was investigated by Movick et al. [158] for the HDO reaction over a ternary TMP, CoPdP, with varying Co/Pd ratios and being supported on ultra-stable Y zeolite (KUSY). The Co-Co, Co-P, Pd-Pd, Pd-P interactions were found by analysis of the X-ray absorption spectroscopy spectra of the catalysts. The binary Pd containing TMP most likely is present as Pd_6_P, while the ternary TMP had higher catalytic activity compared to both the binary ones (Co-P, Pd-P). According to the CO infrared spectroscopy, Co was found to be the active site, whereas TPR studies showed that Pd facilitates Co reduction. 

Along the same lines, dibenzofuran was studied by Molina et al. [159] towards HDO reaction over Ni and Co TMPs. According to the main findings of the study, the Ni_2_P catalyst was found to be a better catalyst than the CoP one, achieving conversion of more than 90% of DBF. Contact time, hydrogen pressure and H_2_/feed ratio were studied. An increase in contact time increased the catalytic activity and the production of oxygen-free products. In particular, at low values of contact time intermediates, such as tetrahydrodibenzofuran (THDBF) and hexahydrodibenzofuran (HHDBF) were found to be formed, while higher contact time gave rise to the formation of mainly bicyclohexane (BCH), proposing the following pathway DBF → HHDBF → THDBF → 2-CHP → BCH. When it comes to H_2_ pressure, both Ni_2_P and CoP catalysts were found to be active in intermediate values of pressure. Finally, the H_2_/DBF ratio is more impactful in the case of CoP rather than the Ni_2_P catalyst. 

### 4.4. On the HDO Activity of the Multi-Elemental TMPs 

In catalysis, a single catalyst is rarely capable of carrying out different types of elementary steps needed to deoxygenate oil-derived compounds. Therefore, bi-functional TMP catalysts have been recently receiving great interest, wherein two different TMPs are employed to provide diverse catalytic active sites for various reaction steps. The coordination of two TMPs phases in a catalytic structure can provide a variety and richer active sites as well as modify the electronic structure of each TMP via interfacial interaction for adjusting their propensity to catalyze different elementary reaction steps within the overall reaction. For instance, Liu and co-workers synthesized heterostructured ${\mathrm{NI}}_{2}P-{\mathrm{NiP}}_{2}$ nanoparticles which performed better than the corresponding single TMP components [160]. Density functional theory calculations showed that a strong charge redistribution took place at the interface, since the average valence charge of P (in ${\mathrm{NiP}}_{2}$), near the interface, decreased from 5.22 to 5.05 eV. Thus, the valence electron state of active catalytic sites could be optimized owing to the existence of hetero-interface. 

Multi-functionality is crucial in catalytic hydrodeoxygenation reactions due to the fact that the HDO catalyst is responsible for multiple reactions such as: C−O bond scission via hydrogenolysis, dissociative adsorption of hydrogen molecule. Mixed phase of TMPs catalysts have exhibited synergistic effect for variety of HDO reactions [161,162,163]. Bonita et al., [161] synthesized a variety of molybdenum-based bimetallic phosphides (MMoP, M = Ru, Ni, Co and Fe), via temperature-programmed reduction (TPR) method, and tested their application in hydrodeoxygenation reaction. The results revealed that charge sharing between the phosphorus and metals controls the relative oxidation of Mo and hence the products distribution. As regards MMoP (M = Fe, Co, Ni), the more oxidized the molybdenum, the higher the selectivity towards benzene from phenol via direct deoxygenation at 400 °C and 750 psig. Cho et al. [163] prepared bimetallic NiFeP catalysts supported on potassium ion-exchanged USY zeolite through TPR method and investigated their activity in HDO reaction of 2-methyltetrahydrofuran (2-MTHF) at temperature range of 250–300 °C. It was found that the NiFeP catalysts comprised of ${\mathrm{Ni}}_{2}P$ and FeP phases as revealed by the XRD results. Notably, the bimetallic NiFeP catalysts presented a weak ligand effect on the reaction rate but a considerable ensemble effect on the products distribution in the HDO of 2-MTHF. Particularly, the NiFeP supported catalysts demonstrated a lower selectivity towards butane as compared to ${\mathrm{Ni}}_{2}P$ supported catalysts, signifying the decarboxylation route is suppressed over the former. 

It should be noted that the controlled synthesis of multi-metallic phosphides remains a challenge because of stringent valence balance demand on the stoichiometric ratio and the phase segregation originating from varying reactivity of various metal precursors [164].

## 5. Deactivation during the HDO Reaction

The main reasons for catalyst deactivation during the HDO reactions are coke formation, metal leaching, sintering, poisoning, and surface oxidation (attack from the water). Coke formation is the main factor of catalyst deactivation since the deposition on the active sites affects the catalyst activity and selectivity when it deposits inside the catalyst pores. The pore size, pore shape, and crystallite size determine the extent and nature of the coke deposit. Carbon is the undesired by-product of polymerization and polycondensation reactions.

The water produced under the HDO reaction conditions can attack P causing its oxidation towards phosphates that may block the active Ni sites and deactivate the catalyst. Additionally, water can induce phase transformation of supports, e.g., converting alumina to boehmite (AlOOH) [165]. According to density functional theory (DFT) calculations, oxygen on Ni_2_P surface prefers to interact with phosphorus. These unphosphide ${\mathrm{Ni}}^{\delta +}$ sites are responsible for phenol production during guaiacol HDO reactions. In addition, the P/Ni ratio of phosphide decreases during the HDO reaction, which is an indication of BrØnsted acid sites (i.e, P-OH) drop on the catalyst surface with a simultaneous increase in the phenol production by suppressing its dehydroxylation. To suppress the coke deposition, a proper adjustment of the operating conditions is needed. For example, hydrogen in the feed saturates the carbon precursor compounds (C_x_H_y_) formed under HDO. In addition, multi-step processes are better/preferred as they assist in minimizing the coking.

Another promising pathway to remove the coke deposits and reduce the deactivation of the catalysts is regeneration. Calcination under oxidant atmosphere at high temperature (500–600 °C) is used to remove the coke deposits. Regeneration cycles reduce the catalyst activity by producing a large amount of acid sites. Regeneration by oxidation is a more efficient reaction method where oxidation and/or reduction (with H_2_) of catalyst. This methodology reduces the cost of biofuel by developing a highly active catalyst.

As reported by Liu et al. [166], O atoms interaction/adsorption with Ni_2_P (001) originating from water can interact with both P and Ni atoms with a strong preference towards P atoms. In the resultant oxy-phosphide species, electron transfer from P to O takes place. Due to the ligand effect in the Ni_2_P, P species may suppress the Ni-O interaction, thus increasing the oxidation resistance of Ni_2_P compared to the Ni/SiO_2_ catalyst. Kelun Li [167], who studied the HDO reaction of anisole over silica-supported Ni2P, MoP, and NiMoP reported that the higher affinity of Mo with oxygen leads to lower oxidation resistance. So, the deactivation of the NiMoP catalyst is decreased as the Ni/Mo ratio is increased.

## 6. Computational Approaches to Study the HDO Reaction over TMPs

Besides the experimental investigation of TMPs catalysts, the Density Function Theory (DFT) calculations have been widely adopted not only for mapping out possible reaction routes at the atomic level but also to deeply scrutinize the rate-controlling steps of reaction pathways. For instance, Wongnongwa and coworkers [168] have systemically studied the deoxygenation mechanisms of palmitic acid over ${\mathrm{Ni}}_{2}P$ catalyst via a multi-pronged approach of experimental, thermodynamics, and DFT studies. The DFT-based computations, based on butyric acid as model molecule of palmitic acid on ${\mathrm{Ni}}_{2}P\;\left(001\right)$ surface, showed that the rate controlling step of decarbonylation (DCO) route is the hydrogenolysis reaction step, i.e., conversion of butanal to propane as shown in Figure 13, whereas that of hydrodeoxygenation (HDO) is the transformation of butanol into butane. The energy barrier of the rate limiting steps for DCO and HDO routes are 1.52 and 1.99 eV, respectively. Considering the decarboxylation (${\mathrm{DCO}}_{2}$) route, the hydrogenolysis step, in which butyric acid is converted to propane, dominates the ${\mathrm{DCO}}_{2}$ reaction, surmounting for a high activation barrier of 2.37 eV. These findings are in line with the experimental observations, which indicated that the DCO pathway is more favorable on ${\mathrm{Ni}}_{2}P$ catalyst than the HDO pathway with a ratio up to 4.13:1, while the ${\mathrm{DCO}}_{2}$ route was found to be unlikely, which was borne out by the absence of the gaseous ${\mathrm{CO}}_{2}$ product. Notably, the thermodynamic analysis indicated that the HDO route becomes more competitive when the high reaction temperature is considered.

It is well-known that the favorable adsorption of reactants and reaction intermediates and their activation could be easily determined by DFT computations, which, in many cases, represent the exact reaction routes during hydrodeoxygenation reactions. For example, Moon et al. [169] (Figure 14) reported that the active site of ${\mathrm{Ni}}_{2}P$ catalyst, for hydrodeoxygenation of guaiacol reaction, consists of P sites and threefold ${\mathrm{Ni}}_{3}$ hollow sites, as confirmed by DFT study together with X-ray absorption fine structure spectroscopy (XAFS) and FTIR measurements, are responsible for adsorption of atomic H or OH groups. These findings indicated that the HDO of guaiacol is greatly affected by the extent of H and OH coverage on the P or Ni sites. In particular, the direct deoxygenation route facilitated the benzene formation on the more OH-covered surface as depicted in Figure 14, while a pre-hydrogenation route is favored on the reduced surface of ${\mathrm{Ni}}_{2}P$ surface.

A fundamental understanding of the reaction mechanism, at the atomistic level, is of prime importance to unravel the rate-limiting steps for the hydrodeoxygenation reaction on TMPs catalysts. Jain et al. [170] investigated the phenol HDO on ternary phosphide catalytic surfaces, namely FeMoP, RuMoP, and NiMoP, using experimental and computational approaches to determine products distribution and C–O bond scission pathways. The results revealed that the FeMoP surface promotes a direct deoxygenation route, thanks to the lower kinetic barrier for cleaving the C-O bond, whereas the NiMoP and RuMoP favor the hydrogenation of the aromatic ring first, followed by the scission of the C-O bond. These results were found to be in accord with product distribution observed from experiments. He et al. [171] performed systematic DFT calculations for investigating the surface chemistry of selected 1st row TM phosphides, particularly VP, TiP, CrP, ${\mathrm{Co}}_{2}P$, ${\mathrm{Ni}}_{2}P$, and ${\mathrm{Fe}}_{2}P$, in the model reaction of phenol deoxygenation. The results revealed that ${\mathrm{Ni}}_{2}P$ and ${\mathrm{Fe}}_{2}P$ exhibited the strongest adsorption energies of phenol, which favored parallel orientation, thanks to the TM ensembles present in these two surfaces. Notably, VP and ${\mathrm{Fe}}_{2}P$ presented lower kinetic barriers for C-O bond cleavage step, as shown in Figure 15, as compared to other the TMPs surfaces. Considering the selective hydrogenation step of phenyl fragment to benzene, the early 1st row TMPs, i.e., VP, CrP and TiP, demonstrated moderate activation energies, whereas lower energy barriers were recorded on ${\mathrm{Ni}}_{2}P$, ${\mathrm{Co}}_{2}P$ and ${\mathrm{Fe}}_{2}P$, pointing out a more facile selective hydrogenation. It is noteworthy that the deoxygenation reaction of phenol towards benzene proceeded via direct scission of $C_{\mathrm{ring}}-\mathrm{OH}$ bond over all TMPs with the exception of TiP, which was revealed to follow indirect route involves the dehydrogenation of ${\mathrm{OH}}_{\mathrm{phenol}}$ then C–O scission as depicted in Figure 15.

It should be noted that TMPs possessed an appropriate level of atomic hydrogen reactivity, carbon affinity, and oxophilicity necessary to efficiently drive recalcitrant phenolic oxygen while maintaining the aromaticity of products. Further insights on how the surface chemistry of TMPs determines their HDO performance have been provided by the work of He and Laursen in the guaiacol deoxygenation over ${\mathrm{Ni}}_{2}P$ and TiP surfaces [172]. It was reported that the activation barrier for the direct demethoxylation step, i.e., removal of ${\mathrm{OCH}}_{3}$ group, is lower on ${\mathrm{Ni}}_{2}P$ (1.0 eV) than TiP (1.4 eV). The dehydroxylation step, which appears to be kinetically hindered in some experimental studies, was reported to be moderately activated with kinetic barriers of 1.3 and 1.2 eV over ${\mathrm{Ni}}_{2}P$ and TiP, respectively. These activation energies signify that TMP are potent catalysts for promoting the removal of both O-containing groups easily under mild process conditions. The parallel orientation of guaiacol on ${\mathrm{Ni}}_{2}P$ surface was reported to inhibit the removal of methoxy group; however, a direct removal of oxygen moieties is thermodynamically favorable when perpendicular geometry is considered. It is well established that the orientation of the aromatic ring plays a decisive role in determining the reaction mechanism since it dictates $C_{\mathrm{ring}}-$ surface interactions, overhydrogenation, C-C bond scission and C=C activation [170]. Parallel geometry can enhance strong $C_{\mathrm{ring}}-$ surface interactions, which modifies the aromatic ring geometry. On the other hand, perpendicular orientation promotes a higher degree of C-O interaction with the catalytic surface and thus more facile cleavage of C-O bond if an adequate surface activity is present. Furthermore, Jia et al. [173] examined the C-OH bond cleavage step of cyclohexanol, being a key oxygenated by-product in guaiacol HDO reaction and precursor of cyclohexane, on ${\mathrm{Ni}}_{3}P_{\;}$ and ${\mathrm{Ni}}_{3}P_{2}$ surface termination of ${\mathrm{Ni}}_{2}P$ crystal. The results revealed that the kinetic barrier of C-OH bond scission on ${\mathrm{Ni}}_{3}P_{\;}$ (1.02 eV) is greater than that of ${\mathrm{Ni}}_{3}P_{2}$ (0.71 eV) terminated surface, signifying the latter could readily break the C-OH bond during HDO reaction of guaiacol. Wang and coworkers [174] studied the anisole HDO reaction mechanism on ${\mathrm{Fe}}_{2}P\;\left(101\right)$ surface, and it was reported that the reaction proceeds via direct scission of $C_{\mathrm{aryl}}-O$ bond of anisole to produce ${\mathrm{CH}}_{3}O^{*}$ and $C_{6}H_{5}^{*}$ reaction intermediate while preventing the formation of $C_{6}H_{5}^{\;}-\mathrm{OH}-{\mathrm{CH}}_{3}$ intermediate, which coincides well with the experimental findings. The results revealed that $C_{\mathrm{aryl}}-O$ bond of anisole could be readily cleaved, requiring an energy barrier of 0.75 eV. The activation barriers were also reported for hydrogenation of $C_{6}H_{5}^{*}$ (0.08 eV) and ${\mathrm{CH}}_{3}O^{*}$ (1.10 eV), to form benzene and methanol, respectively. These findings suggest that the formation of the methanol reaction step dominates the HDO reaction on the ${\mathrm{Fe}}_{2}P\;\left(101\right)$ surface.

A few computational studies have reported the deoxygenation of furan derivatives on TMPs catalysts [174,175] For instance, Witzke et al. [175] investigated the hydrogenolysis mechanism of 2-Methyltetrahydrofuran (MTHF) over ${\mathrm{Ni}}_{2}P$(001), ${\mathrm{Ni}}_{12}P_{5}$(001), and Ni(111) catalysts. DFT computations revealed that increasing the P:Ni ratio of the catalyst have resulted in decreasing the activation barrier for C 2 −O bond rupture, pertained to conversion of MTHF to 2-pentene-4-one intermediate pathway, as compared to that of C 3 −O bond rupture related to hydrogenolysis pathway involving the formation of pentanal intermediate. The exposed ${\mathrm{Ni}}_{3}$ hollow sites, present on ${\mathrm{Ni}}_{2}P$ and Ni surfaces, and ${\mathrm{Ni}}_{4}$ hollow sites, existed on ${\mathrm{Ni}}_{12}P_{5}$ surface, are the active sites for cleaving C  −O bond.

Regarding the sintering and coking, surface oxidation is deemed to be a possible source of catalyst deactivation [176,177]. Cecilia and coworkers [177] reported that phosphorus in ${\mathrm{Ni}}_{2}P$ catalyst could be oxidized, during the course of the reaction, to form phosphate, which is expected to cover the metallic active sites in TMPs catalysts, thereby resulting in catalyst deactivation. First-principles calculations revealed that atomic oxygen on a ${\mathrm{Ni}}_{2}P$ surface interact preferably with P surface atoms [178], which would be the convenient environment for oxidizing phosphorus to undesired phosphate species. It can be concluded that there is a pressing need for more computational studies to be considered for deactivation and regenerations of TMPs catalysts and HDO catalytic mechanisms at the atomistic level since these aspects are crucial before applying technology into industrial manufacture of biofuels. In this context, DFT computations should be adopted to deeply identify the detailed pathways and the rate-controlling steps for the HDO process.

Rensel et al. [178] studied the phenol HDO reaction on bimetallic ${\mathrm{Fe}}_{x}{\mathrm{Mo}}_{2-x}P$ (x = 1 and 1.5) catalysts using both experimental and DFT computational approaches to elucidate the role of surface Lewis acidity on the reaction mechanism. The results indicated that the rate determining step is the $C_{\mathrm{ring}}-O$ scission because it has the greatest activation barrier among the studied elementary reaction steps. The $C_{\mathrm{ring}}-O$ bond cleavage takes place readily on ${\mathrm{Fe}}_{1}{\mathrm{Mo}}_{1}P$ than ${\mathrm{Fe}}_{1.5}{\mathrm{Mo}}_{0.5}P$ surface, surmounting for the kinetic barrier of 0.39 eV and 0.77 eV, respectively, which was explained by the greater partial charge on metallic species of the former (+0.81 | *e* |) as compared to the latter (+0.63 | *e* |). This can be correlated to the improved Lewis acid character of the ${\mathrm{Fe}}_{1}{\mathrm{Mo}}_{1}P$ as compared to ${\mathrm{Fe}}_{1.5}{\mathrm{Mo}}_{0.5}P$ catalyst.

Wanmolee and co-workers [133] studied the HDO of guaiacol on ${\mathrm{Cu}}_{3}P$/${\mathrm{SiO}}_{2}$ catalysts. The most predominant facets based on XRD analysis, the guaiacol adsorption was investigated on (100) and (113) surfaces of ${\mathrm{Cu}}_{3}P$. It was reported that the active sites for adsorbing the reactants, guaiacol and $H_{2}$, are Cu sites for the studied surfaces. The adsorption of guaiacol is more favorable via Cu-OH and Cu-C binding modes than Cu-${\mathrm{OCH}}_{3}$ mode because of the steric hindrance of the latter. The ${\mathrm{Cu}}_{3}P$ (113) surface presented an enhanced adsorption capability as compared to (100) for all reactants and reaction intermediates, which was correlated with the lesser stability of the former as confirmed by surface energy results. The Bader charge analysis revealed that the P sites are negatively charged while the Cu sites are positively charged, acting as Lewis acid sites.

## 7. Future Perspectives and Outlook

The up-to-date advancements of the HDO catalytic chemistry for converting pyrolysis bio-oils to fuels show that the process needs more time to be considered mature. TMPs are only one class of catalysts being successfully used for this reaction and with clear benefits compared to noble metal or pure transition metal catalysts. The complex electronic structure along with the bulk and surface properties of TMPs allow for a bifunctional catalytic chemistry to take place while it can be tuned by adding second metal or support. The increased activity of the TMPs towards oxygen drives the C–O bond breaking. The different electronic structure of the early TMP and the different extent of hybridization gives room for the rational design of catalysts. The above can be tailored by choosing the appropriate M, P precursors, and synthesis method to control the morphology and the size of the final TMP. The P precursors can be challenging as many are available (organic and inorganic). The role of P sites is often limited to that of spectator or binding provider. Paying more attention to the active site and the role of P in formulating that through a mechanistic approach would benefit the rational design of catalysts. In such a case, a better understanding will improve the TMPs stability for real life conditions/applications. In that direction, it would be worth understanding the TMPs behavior under other reactions/conditions where the reactants, temperature, pressure vary. Having a closer look at the different reaction steps, the breaking of C–C/C–O bonds needs to be tailored as it leads to the loss of atoms in biomass; from that aspect, an atom-economical approach needs to be followed. In such an attempt, the OCH_3_ (methoxy) groups can be used for chemical production so to utilize the carbon atoms efficiently. Additionally, OCH_3_/OH groups can be used as H_2_ carriers to self-support the HDO reaction, while dealing with the water produced during HDO is worth more attention as the latter can facilitate structure modifications or can co-catalyze some reactions steps (e.g., breaking of C–O bond). Another aspect is the facet-controlled growth of the TMPs and the study of the different exposed facets. Succeeding to grow TMPs that expose their most catalytically active facet can further add to the design of a catalyst. When it comes to the multi-elemental TMPs, such as ternary TMPs, strict control of the preparation conditions is needed (pH, P source, temperature, complexing agent) as in these systems, the activity of two metals is combined with P and thus control of the final stoichiometry is a challenging task. Multi-elemental TMPs catalysts have been rarely studied for HDO reactions. The key reason behind the importance of multi-elemental TMPs is the necessity of both hydrogenation and base/acid active sites. While many multi-elemental TMPs catalysts presented promising activity in HDO reactions, some key unresolved questions regarding the mechanism remain. Particularly, there is a pressing need to unravel the role of the interface between the different TMP phases in the reaction mechanism. In most studies to date, the density of interfacial sites has not been explicitly controlled, so it is difficult to discern the role proximity between distinct types of active sites plays in the reaction mechanism. It is importance to resolve these questions since the rational design of multi-elemental TMPs catalysts would consider tailoring the interface between the different TMPs phases. When these key challenges of TMPs catalysts can be overcome in the next decade, we foresee a promising potential for the industrial application of TMPs and multi-elemental TMPs for converting pyrolysis bio-oils to fuels. When it comes to the HDO reaction itself and how the plethora of TMP compositions impact the activity, more insightful studies are needed for the different process parameters, such as H_2_ pressure, H_2_/feed ratio, contact time, and how these parameters impact, from a mechanistic point of view the products selectivity/reaction pathway.

Additionally, the development of a cost-effective and easily scalable synthesis protocol still remains one of the barriers to the commercial applicability of TMPs and multi-elemental TMPs and catalysts. Although some of the discussed TMPs catalysts exhibited promising bi-functional catalytic activity for HDO reactions, most of the reported preparation methods are still restricted to lab scale. Therefore, developing a scalable and abundant preparation strategy plays an essential role in the realization of the industrial application of TMPs catalysts.

## Figures and Tables

**Figure 1 nanomaterials-12-01435-f001:**
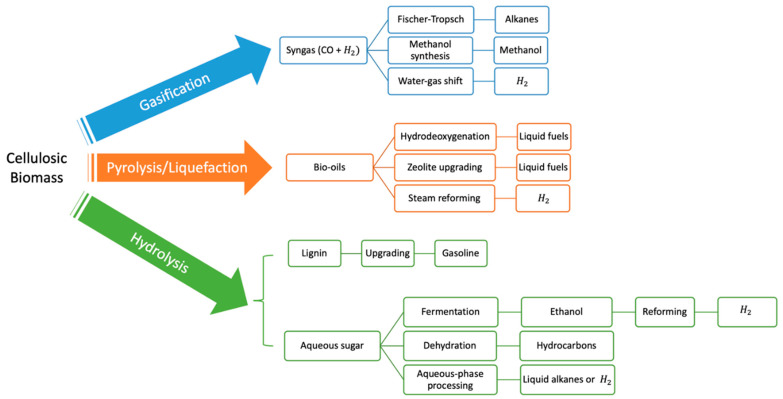
Process for the fuel production starting from cellulosic biomass.

**Figure 2 nanomaterials-12-01435-f002:**
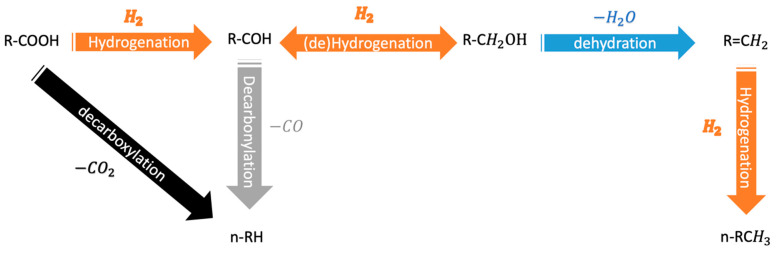
Hydrodeoxygenation reaction network.

**Figure 3 nanomaterials-12-01435-f003:**
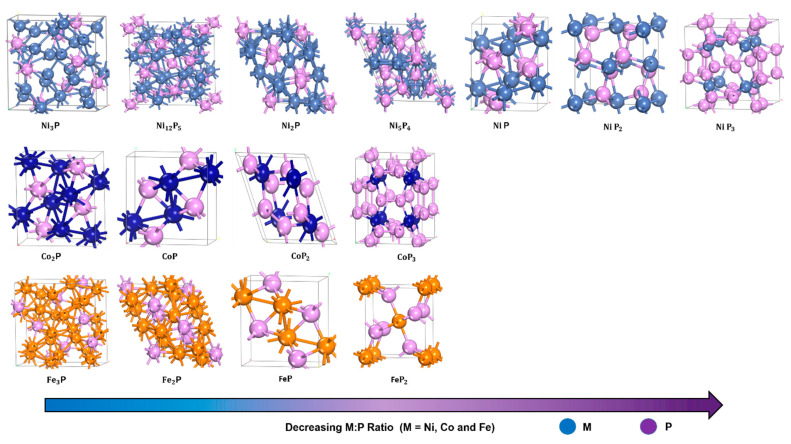
Versatility of Crystal Structures of TMPs.

**Figure 4 nanomaterials-12-01435-f004:**
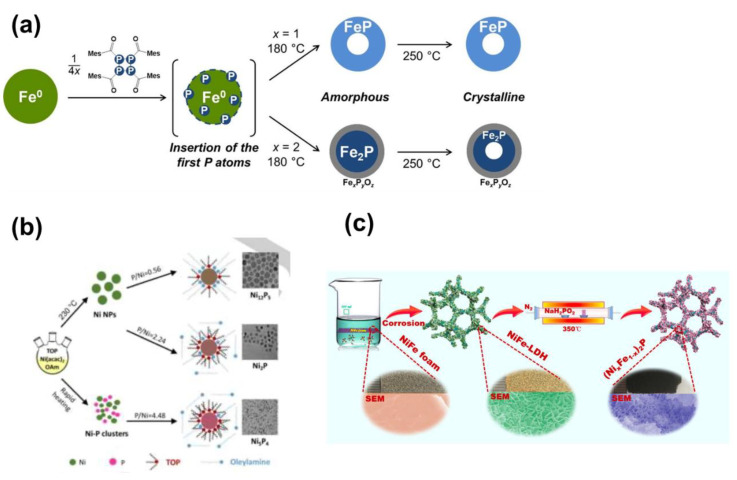
(**a**) Fe-P synthesis. Reprinted with permission from Ref. [97], Copyright 2020 Wily Online; (**b**) NiP nanocrystals formation. Reprinted with permission from Ref. [98], Copyright 2018 Wily Online; (**c**) Synthesis of (Ni_x_Fe_1−x_)_2_P nanosheets via phosphorization of the NiFe-LDH precursors. Reprinted with permission from Ref. [99], Copyright 2020 ACS Publication.

**Figure 5 nanomaterials-12-01435-f005:**
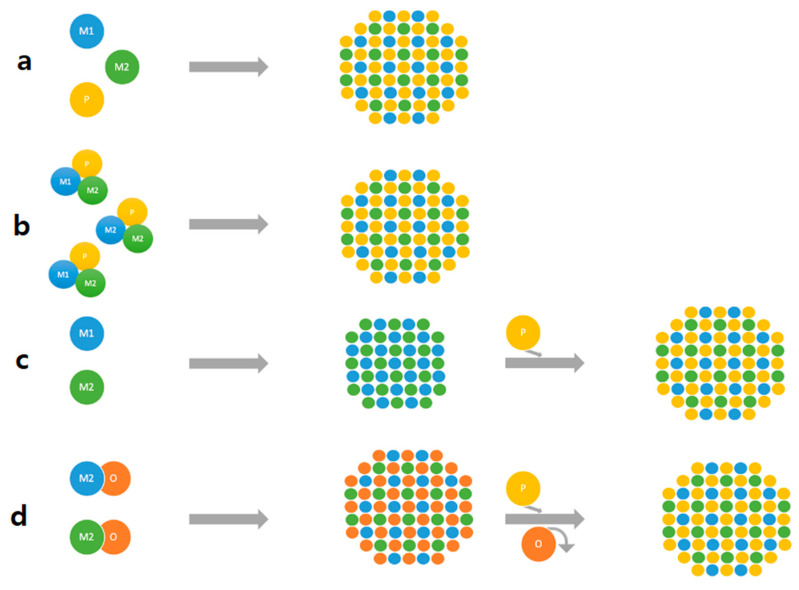
Colloidal synthesis of ternary TMPs. Adapted from Ref [41], (**a**) reaction of a phosphorus precursor (P) with both transition metal sources (M1 and M2), (**b**) Thermal decomposition of a single precursor consisting of phosphorus and both metals, (**c**) generation of bi-metallic nanoparticles their subsequent phosphorisation reaction, (**d**) generation of oxidic nanoparticles and their subsequent phosphorisation.

**Figure 6 nanomaterials-12-01435-f006:**
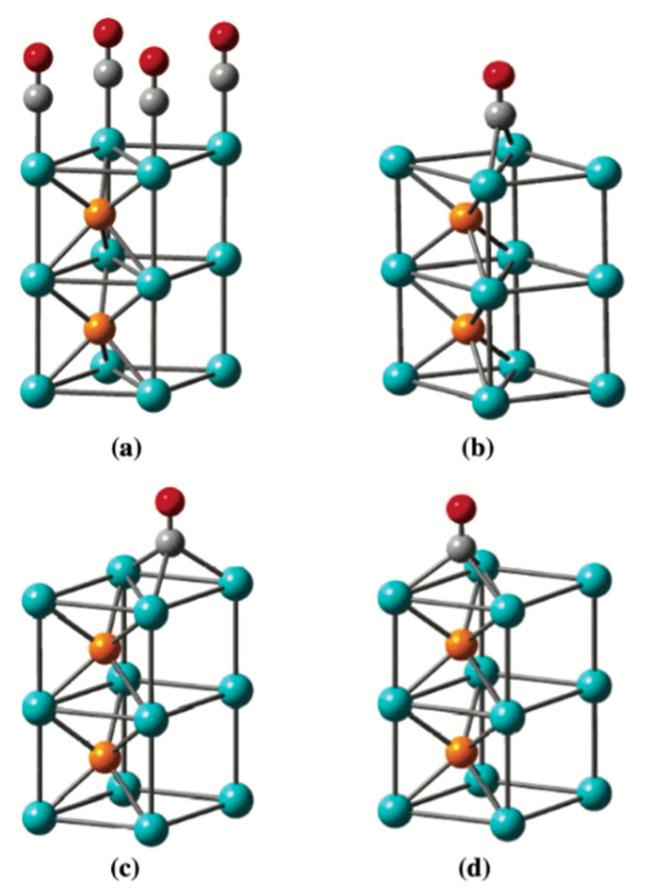
CO chemisorption on MoP-(001)-(1 × 1) surface: (**a**) on-top Mo, (**b**) bridge, (**c**) fcc site, and (**d**) hcp hollow site. Reprinted with permission from Ref. [107]. Copyright 2017 Springer Nature.

**Figure 7 nanomaterials-12-01435-f007:**
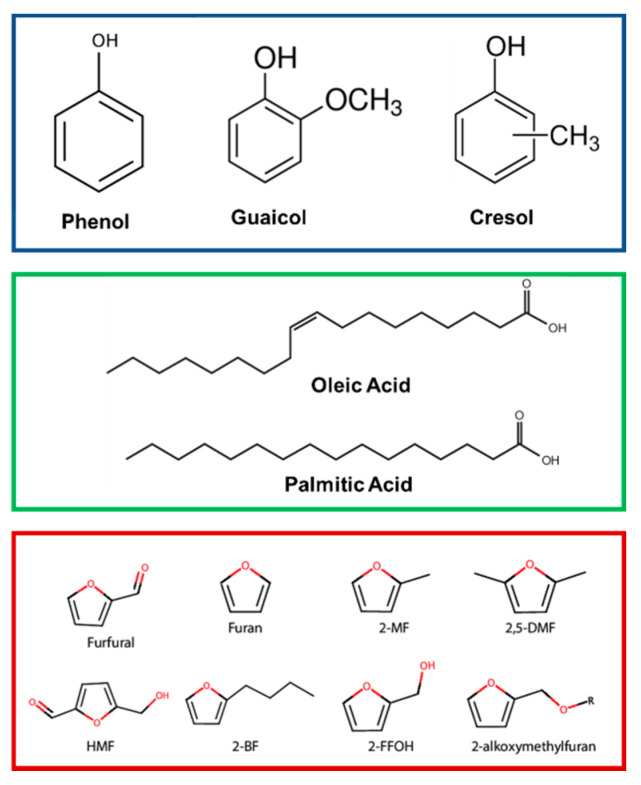
Representative compounds from lignin phenols, fatty acids and cellulose furans.

**Figure 8 nanomaterials-12-01435-f008:**
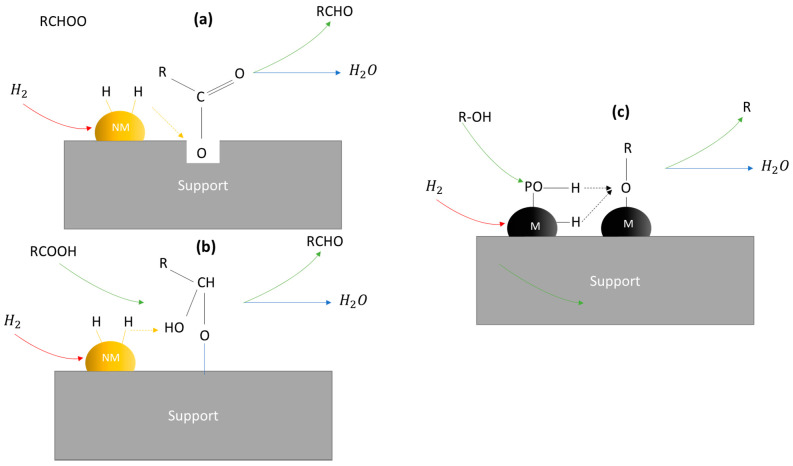
(**a**,**b**) HDO mechanisms over noble metal (NM) supported catalysts. Adapted from Refs [135,136]. (**c**) Proposed HDO mechanisms over TMPs catalysts. Adapted from Refs [137,138], M stands for metals.

**Figure 9 nanomaterials-12-01435-f009:**
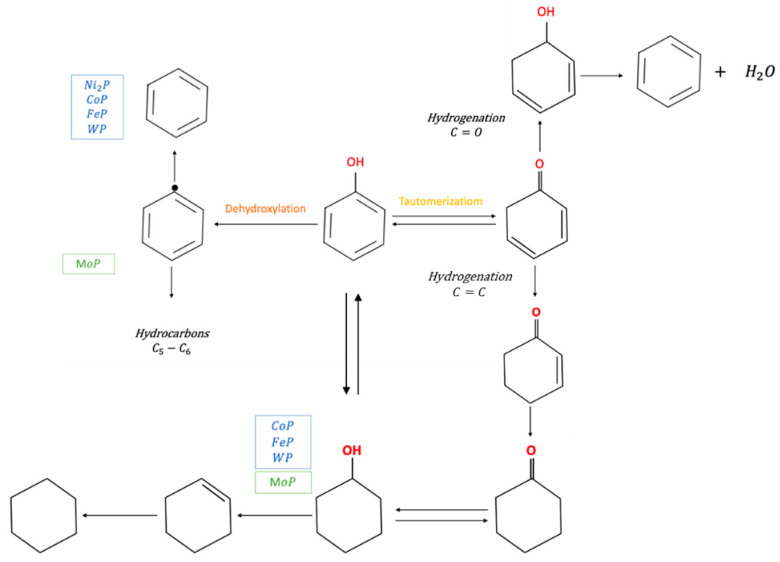
Reaction pathway for HDO of phenol over various supported TMP catalysts. Adapted from Ref. [118].

**Figure 10 nanomaterials-12-01435-f010:**
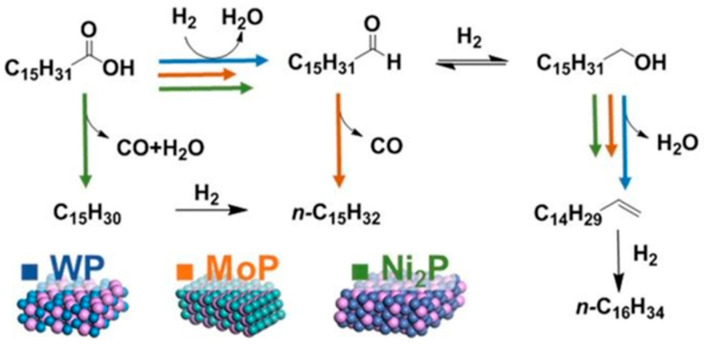
Different mechanistic pathways of the reaction. Reprinted with permission from Ref. [151]. Copyright 2017 ACS Publication.

**Figure 11 nanomaterials-12-01435-f011:**
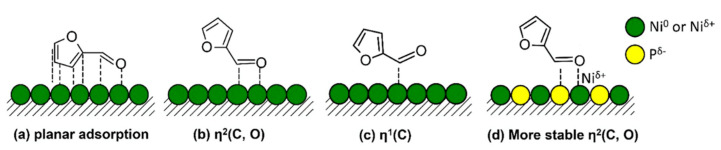
Adsorption configurations of furfural: (**a**) planar adsorption on Ni surface, (**b**) furfural adsorbed through its C and carbonyl O atoms on Ni surface, (**c**) furfural coordinated to Ni surface via C atom on Ni surface and (**d**) furfural adsorption via C and carbonyl O atoms on NiP(x) surface. Reprinted with permission from Ref. [98]. Copyright 2018 Wily Online.

**Figure 12 nanomaterials-12-01435-f012:**
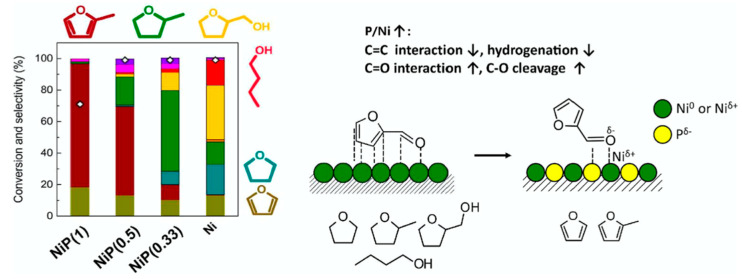
Effect of P on furfural HDO over Ni and nickel phosphide catalysts. Reprinted with permission from Ref. [98]. Copyright 2018 Wily Online.

**Figure 13 nanomaterials-12-01435-f013:**
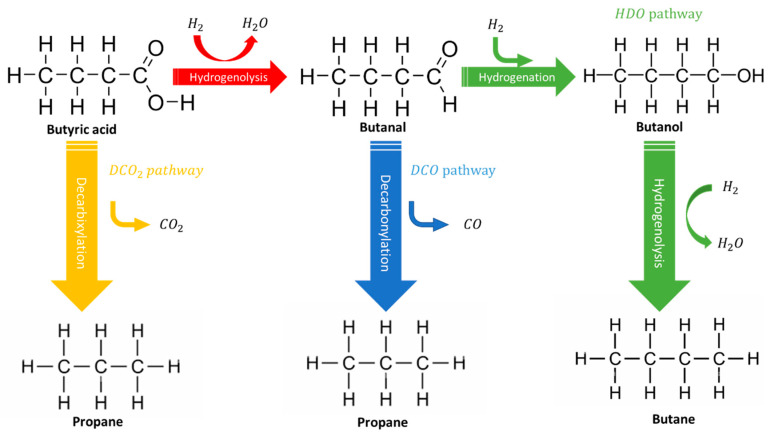
The proposed decarboxylation (DCO_2_), decarbonylation (DCO) and hydrodeoxygenation (HDO) mechanisms for deoxygenation process. Adapted from Ref. [168].

**Figure 14 nanomaterials-12-01435-f014:**
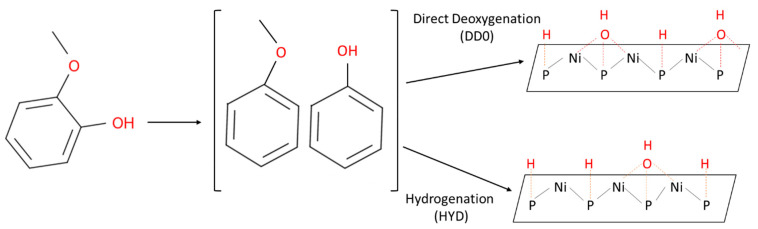
Suggested reaction pathways for guaiacol HDO on ${\mathrm{Ni}}_{2}P$ catalyst. Adapted from Ref. [169].

**Figure 15 nanomaterials-12-01435-f015:**
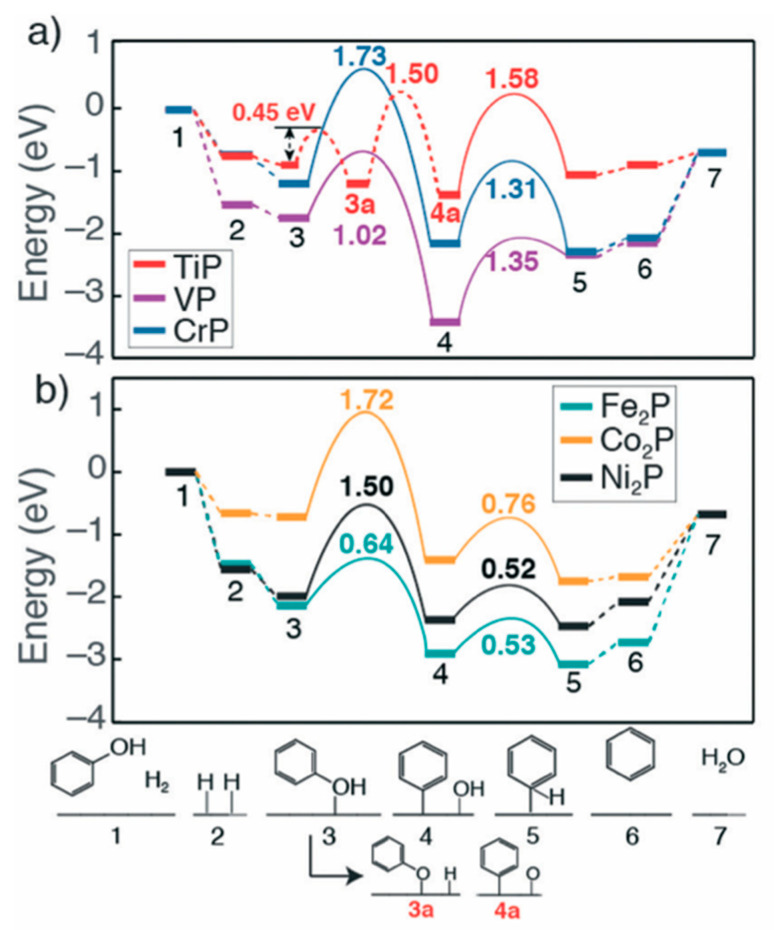
Potential energy landscape for phenol deoxygenation reaction over (**a**) TiP, VP and CrP and (**b**) Fe_2_P, Co_2_P and Ni_2_P surfaces. Reprinted with permission from Ref. [171]. Copyright 2017 ACS Publication.

**Table 1 nanomaterials-12-01435-t001:** Categorization of metal phosphides.

** Metal Phosphide Classes **	Examples of Metal Phosphides
Metal-rich phosphides	$M_{x}P_{y}\left(x>y\right);{\;\mathrm{Ni}}_{2}P,{\mathrm{Co}}_{2}P,{\;\mathrm{Ni}}_{3}P,\;\mathrm{etc}.$
Monophosphides	$M_{x}P_{y}\left(x=y\right);\;\mathrm{MoP},\mathrm{WP},\;\mathrm{CoP},\;\mathrm{etc}.$
Phosphorous-rich phosphides	$M_{x}P_{y}\left(x<y\right);{\;\mathrm{NiP}}_{2},{\mathrm{FeP}}_{2},\;\mathrm{etc}.$
Ionic phosphides	$M_{x}^{n+}P_{y}^{n-};{\;\mathrm{Th}}_{3}P_{4},\;\mathrm{etc}.$

**Table 2 nanomaterials-12-01435-t002:** Physical properties of metal-rich phosphides.

Ceramic Properties			Metallic Properties		
Melting point	°C	830–1530	Electrical resistivity	μΩcm	900–25,000
Microhardness	$\mathrm{kg}\cdot {\mathrm{mm}}^{-2}$	600–1100	Magnetic susceptibility	$10^{6}{\mathrm{cm}}^{3}\cdot {\mathrm{mol}}^{-1}$	110–620
Heat of formation	$\mathrm{kJ}\cdot {\mathrm{mol}}^{-2}$	30–180	Heat capacity	$J\cdot {\left(\mathrm{mol}\cdot {}^{\circ}C\right)}^{-1}$	20–50

**Table 3 nanomaterials-12-01435-t003:** Different TMPs and their crystalline structure.

Phosphide Phase	Crystal System, Space Group	Number of Formula Units	References
${\mathrm{Ni}}_{2}P$	Hexagonal $P\bar{6}2m$ Ni/P=2	3	[1,2,3,4,5,6,7,8,9,10,11,12,13,14,15,16,17,18,19,20,21,22,23,24,25,26,27,28,29,30,31,32,33,34,35,36,37,38,39,40]
${\mathrm{Ni}}_{12}P_{5}$	Tetragonal $\bar{I4}m$ Ni/P=2.4	2	[1,2,3,4,5,6,7,8,9,10,11,12,13,14,15,16,17,18,19,20,21,22,23,24,25,26,27,28,29,30,31,32,33,34,35,36,37,38,39,40]
${\mathrm{Ni}}_{3}P$	Tetragonal $\bar{I4}$ Ni/P=3	8	[1,2,3,4,5,6,7,8,9,10,11,12,13,14,15,16,17,18,19,20,21,22,23,24,25,26,27,28,29,30,31,32,33,34,35,36,37,38,39,40]
${\mathrm{Ni}}_{5}P_{4}$	Hexagonal $P6_{3}\mathrm{mc}$ Ni/P=1.25	4	[1,2,3,4,5,6,7,8,9,10,11,12,13,14,15,16,17,18,19,20,21,22,23,24,25,26,27,28,29,30,31,32,33,34,35,36,37,38,39,40]
CoP	Orthorhombic Pnma Co/P=1	4	JCPDS No.29-0497
${\mathrm{Co}}_{2}P$	Orthorhombic Pnma Co/P=2	4	JCPDS No.89-3030
MoP	Hexagonal $P\bar{6}2m$ Mo/P=1	1	[1,2,3,4,5,6,7,8,9,10,11,12,13,14,15,16,17,18,19,20,21,22,23,24,25,26,27,28,29,30,31,32,33,34,35,36,37,38,39,40]
${\mathrm{Mo}}_{3}P$	Tetragonal $\bar{I4}2m$ Mo/P=3	8	[1,2,3,4,5,6,7,8,9,10,11,12,13,14,15,16,17,18,19,20,21,22,23,24,25,26,27,28,29,30,31,32,33,34,35,36,37,38,39,40]
WP	Orthorhombic Pnma W/P=1	4	JCPDS No.29-1364

## Data Availability

Not Applicable.

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
