# Peer review of "Transition Metal Phosphides (TMP) as a Versatile Class of Catalysts for the Hydrodeoxygenation Reaction (HDO) of Oil-Derived Compounds"

_nanomaterials, 2022, doi:10.3390/nano12091435_

Round 1

Reviewer 1 Report

The authors focus on the unique structural and electronic properties of the transition metal phosphides (TMPs) and their reactivity towards the HDO reaction. I am not sure about the main novelty of this paper, and how this paper distinguishes itself from previous review paper? This manuscript needs minor revision and I have listed these issues and recommendations in chronological order. Following is a summary of the major corrections and revisions:

1. The authors may need to briefly address the difference(s) between the current manuscript and other similar published review articles in the Introduction section.

2. In general, the references in the introduction are poorly chosen, when compared to the sentences they serve as confirmation for.

3. Conclusions: Conclusions need to be improved by specifying the discussed important points within this work. In the conclusions, the authors should also provide an outlook of the challenges and potential future directions.

Author Response

Response to the Reviewers

We would like to express our sincere gratitude for the constructive comments we received from the reviewers. In an effort to address, to the best of our knowledge, the reviewers’ comments, several changes have been made to the manuscript. All the changes are given in the revised manuscript highlighed in green.  Below is a point-by-point response to the reviewer’s comments in blue font.

Reviewer #1:         

The authors focus on the unique structural and electronic properties of the transition metal phosphides (TMPs) and their reactivity towards the HDO reaction. I am not sure about the main novelty of this paper, and how this paper distinguishes itself from previous review paper? This manuscript needs minor revision and I have listed these issues and recommendations in chronological order. Following is a summary of the major corrections and revisions:

Comment #1: The authors may need to briefly address the difference(s) between the current manuscript and other similar published review articles in the Introduction section.

Response: We thank the reviewer for this valuable comment. This manuscript mainly focuses on HDO reaction using transition metal phosphides. The review article touches upon both the experimental and computational studies available in the literature on the TMPs as catalysts for the HDO reaction. In the experimental front, this review article covers all aspects of HDO reaction, as well as the TMPs synthesis, and their characterization with sophisticated techniques. TMPs particular structural characteristics (e.g. single TMPs, binary TMPs, faceted-growth of TMPs) and how this make TMPs a potential class of catalysts for the upgrading of bio-oil is also discussed.  Furthermore, this review discusses the deactivation aspects of TMPs during the course of the reaction. Additionally, this manuscript addresses various kinds of bio-oil feedstocks derived from the lignin-derivatives, cellulose derivatives and fatty acids, such as phenols, furans and fatty acids. The span of the topics discussed in this review cannot usually be found in one individual review article as usually review articles are focused on one of the following topics; synthesis of TMPs, catalysis of different compounds and deactivation. 

Comment #2: In general, the references in the introduction are poorly chosen, when compared to the sentences they serve as confirmation for.

Response:  We thank the reviewer for this comment. The poor references in the introduction section have now been enriched with new references.

Comment #3: Conclusions: Conclusions need to be improved by specifying the discussed important points within this work. In the conclusions, the authors should also provide an outlook of the challenges and potential future directions.

Response: Some challenges pertaining to TMPs and multi-elemental TMPs are mentioned, and the below discussion was included in the manuscript: (Please check page 43/Section 7).

Reviewer 2 Report

The review is focused on the unique structural and electronic properties of the transition metal phosphides (TMP) and their reactivity towards the HDO reaction. Although a wide review of these processes has been carried out in the proposed manuscript there are some lacks which needs to be solved before being considered for publication.

1. The species of transition metals require different catalytic steps and the use of mixed phase of TMP catalysts to make them more efficient is an important challenge for the future. Although, this issue has been commented in the summary, several examples have been described in literature and it must be included and commented in the review. Probably, these works are one the main achievements in this field in the last years and an important challenge for the next years.

2. The reaction mechanisms over nickel phosphide catalysts are already well-known and some have dedicated reviews. The author must focus on another TMPs.

3. The catalytic reactions from Table 4 are not interesting and their information relates to obvious reports. I could not learn anything new from the tables.

4. More focus in this issue is necessary in the review especially the roles of supports on reducibility and activities of TMP catalysts for HDO are required to review.

5. In situ XAS, especially the XANES analysis on phase transformation of metal phosphates to metal phosphides are required to add (section 3.3)

6. Reference number 102 needs to replace (or should add doi number)

Note.

7. Some typos and errors are found in the whole manuscript

Author Response

Response to the Reviewers

We would like to express our sincere gratitude for the constructive comments we received from the reviewers. In an effort to address, to the best of our knowledge, the reviewers’ comments, several changes have been made to the manuscript. All the changes are given in the revised manuscript highlighed in green.  Below is a point-by-point response to the reviewer’s comments in blue font.

Reviewer #2:         

The review is focused on the unique structural and electronic properties of the transition metal phosphides (TMP) and their reactivity towards the HDO reaction. Although a wide review of these processes has been carried out in the proposed manuscript there are some lacks which needs to be solved before being considered for publication.

Comment #1: The species of transition metals require different catalytic steps and the use of mixed phase of TMP catalysts to make them more efficient is an important challenge for the future. Although, this issue has been commented in the summary, several examples have been described in literature and it must be included and commented in the review. Probably, these works are one the main achievements in this field in the last years and an important challenge for the next years.

Response: We thank the reviewer for this constructive comment and we agree for his/her suggestion to add examples and discussion on the multi-phase TMPs. The use of bi-functional TMP catalysts in HDO reactions and the challenges associated with their synthesis are discussed in detail. A new section is now created in the revised manuscript (Please check Section 4.4).

Comment #2: The reaction mechanisms over nickel phosphide catalysts are already well-known and some have dedicated reviews. The author must focus on another TMPs.

Response: We do agree with the reviewer. Bimetallic  and   were also considered and the below discussion was included in the manuscript: (Please check page 42/Section 6).

Comment #3: The catalytic reactions from Table 4 are not interesting and their information relates to obvious reports. I could not learn anything new from the tables.

Response: Table 4 was replaced with a new table that is dedicated to HDO reactions only for various oil-derived compounds. The reaction conditions and process metrics (conversion, selectivity) are now also included in this table so to be more instructive.

Comment #4: More focus in this issue is necessary in the review especially the roles of supports on reducibility and activities of TMP catalysts for HDO are required to review.

Response: We thank the reviewer for shedding light into the importance of supports on the catalytic activity, the role of supports on activity and formation of metal phosphides phase was addressed accordingly and the below discussion was included in the manuscript: (Please check page 10/Section 2.1).

Comment #5: In situ XAS, especially the XANES analysis on phase transformation of metal phosphates to metal phosphides are required to add (section 3.3)

Response: We would like to thank the reviewer for this suggestion as it allowed us to strengthen the discussion. Therefore, in situ XAS analysis, particularly EXAFS and XANES, was added and the below elaborate discussion was included in the manuscript (Please check page 24/Section 3.3):

Comment #6: Reference number 102 needs to replace (or should add doi number)

Response: https://doi.org/10.1016/j.cattod.2020.07.077. References number has been changed as new references have bee added.

Comment #7: Note. Some typos and errors are found in the whole manuscript

Response:  Below are some typos amended in the revised manuscript:

Phospites

Incontact

Inveigated

There are two figures having the caption of figure 10. Same goes for Table 3

Figure 4-b is taken from ref 85 not 24

Figure 5 is taken from ref 34 not 93

Round 2

Reviewer 2 Report

The authors have addressed all my point very well.